# TRAINING GANS WITH STRONGER AUGMENTATIONS VIA CONTRASTIVE DISCRIMINATOR

**Jongheon Jeong**[1] **& Jinwoo Shin**[2,1]
[1]School of Electrical Engineering  [2]Graduate School of AI
Korea Advanced Institute of Science and Technology (KAIST)
Daejeon 34141, South Korea
{jongheonj,jinwoos}@kaist.ac.kr

## ABSTRACT

Recent works in Generative Adversarial Networks (GANs) are actively revisiting various data augmentation techniques as an effective way to prevent discriminator overfitting. It is still unclear, however, that which augmentations could actually improve GANs, and in particular, how to apply a wider range of augmentations in training. In this paper, we propose a novel way to address these questions by incorporating a recent *contrastive representation learning* scheme into the GAN discriminator, coined *ContraD*. This "fusion" enables the discriminators to work with much stronger augmentations without increasing their training instability, thereby preventing the discriminator overfitting issue in GANs more effectively. Even better, we observe that the contrastive learning itself *also* benefits from our GAN training, *i.e.*, by maintaining discriminative features between real and fake samples, suggesting a strong coherence between the two worlds: good contrastive representations are also good for GAN discriminators, and *vice versa*. Our experimental results show that GANs with ContraD consistently improve FID and IS compared to other recent techniques incorporating data augmentations, still maintaining highly discriminative features in the discriminator in terms of the linear evaluation. Finally, as a byproduct, we also show that our GANs trained in an unsupervised manner (without labels) can induce many conditional generative models via a simple latent sampling, leveraging the learned features of ContraD. Code is available at https://github.com/jh-jeong/ContraD.

## 1 INTRODUCTION

Generative adversarial networks (GANs) (Goodfellow et al., 2014) have become one of the most prominent approaches for generative modeling with a wide range of applications (Ho & Ermon, 2016; Zhu et al., 2017; Karras et al., 2019; Rott Shaham et al., 2019). In general, a GAN is defined by a minimax game between two neural networks: a *generator* network that maps a random vector into the data domain, and a *discriminator* network that classifies whether a given sample is real (from the training dataset) or fake (from the generator). Provided that both generator and discriminator attain their optima at each minimax objective alternatively, it is theoretically guaranteed that the generator implicitly converges to model the data generating distribution (Goodfellow et al., 2014).

Due to the non-convex/stationary nature of the minimax game, however, training GANs in practice is often very unstable with an extreme sensitivity to many hyperparameters (Salimans et al., 2016; Lucic et al., 2018; Kurach et al., 2019). Stabilizing the GAN dynamics has been extensively studied in the literature (Arjovsky et al., 2017; Gulrajani et al., 2017; Miyato et al., 2018; Wei et al., 2018; Jolicoeur-Martineau, 2019; Chen et al., 2019; Schonfeld et al., 2020), and the idea of incorporating *data augmentation* techniques has recently gained a particular attention on this line of research: more specifically, Zhang et al. (2020) have shown that *consistency regularization* between discriminator outputs of clean and augmented samples could greatly stabilize GAN training, and Zhao et al. (2020c) further improved this idea. The question of which augmentations are good for GANs has been investigated very recently in several works (Zhao et al., 2020d; Tran et al., 2021; Karras et al., 2020a; Zhao et al., 2020a), while they unanimously conclude only a limited range of augmentations (*e.g.*, flipping and spatial translation) were actually helpful for the current form of training GANs.

Meanwhile, not only for GANs, data augmentation has also been played a key role in the literature of self-supervised representation learning (Doersch et al., 2015; Gidaris et al., 2018; Wu et al., 2018), especially with the recent advances in *contrastive learning* (Bachman et al., 2019; Oord et al., 2018; Chen et al., 2020a;b; Grill et al., 2020): *e.g.*, Chen et al. (2020a) have shown that the performance gap between supervised- and unsupervised learning can be significantly closed with large-scale contrastive learning over strong data augmentations. In this case, contrastive learning aims to extract the mutual information shared across augmentations, so good augmentations for contrastive learning should keep information relevant to downstream tasks (*e.g.*, classification), while discarding nuisances for generalization. Finding such augmentations is still challenging, yet in some sense, it is more tangible than the case of GANs, as there are some known ways to formulate the goal rigorously, *e.g.*, InfoMax (Linsker, 1988) or InfoMin principles (Tian et al., 2020).

**Contribution.** In this paper, we propose *Contrastive Discriminator* (ContraD), a new way of training discriminators of GAN that incorporates the principle of contrastive learning. Specifically, instead of directly optimizing the discriminator network for the GAN loss, ContraD uses the network mainly to extract a *contrastive representation* from a given set of data augmentations and (real or generated) samples. The actual discriminator that minimizes the GAN loss is defined independently upon the contrastive representation, which turns out that a simple 2-layer network is sufficient to work as a complete GAN. By design, ContraD can be naturally trained with augmentations used in the literature of contrastive learning, *e.g.*, those proposed by SimCLR (Chen et al., 2020a), which are in fact much stronger than typical practices in the context of GAN training (Zhang et al., 2020; Zhao et al., 2020c;a; Karras et al., 2020a). Our key observation here is that, the task of contrastive learning (to discriminate each of independent real samples) and that of GAN discriminator (to discriminate fake samples from the reals) benefit each other when jointly trained with a shared representation.

Self-supervised learning, including contrastive learning, have been recently applied in GAN as an auxiliary task upon the GAN loss (Chen et al., 2019; Tran et al., 2019; Lee et al., 2021; Zhao et al., 2020d), mainly in attempt to alleviate catastopic forgetting in discriminators (Chen et al., 2019). For conditional GANs, Kang & Park (2020) have proposed a contrastive form of loss to efficiently incorporate a given conditional information into discriminators. Our work can be differentiated to these prior works in a sense that, to the best of our knowledge, it is the first method that successfully leverage contrastive learning alone to incorporate a wide range of data augmentations in GAN training. Indeed, for example, Zhao et al. (2020d) recently reported that simply regularizing auxiliary SimCLR loss (Chen et al., 2020a) improves GAN training, but could not outperform existing methods based on simple data augmentations, *e.g.*, bCR (Zhao et al., 2020c).

## 2 BACKGROUND

**Generative adversarial networks.** We consider a problem of learning a generative model $p_g$ from a given dataset $\{\boldsymbol{x}_i\}_{i=1}^N$, where $\boldsymbol{x}_i \sim p_{\text{data}}$ and $\boldsymbol{x}_i \in \mathcal{X}$. To this end, *generative adversarial network* (GAN) (Goodfellow et al., 2014) considers two neural networks: (a) a *generator* network $G : \mathcal{Z} \to \mathcal{X}$ that maps a latent variable $\mathbf{z} \sim p(\mathbf{z})$ into $\mathcal{X}$, where $p(\mathbf{z})$ is a specific prior distribution, and (b) a *discriminator* network $D : \mathcal{X} \to [0, 1]$ that discriminates samples from $p_{\text{data}}$ and those from the implicit distribution $p_g$ derived from $G(\mathbf{z})$. The primitive form of training $G$ and $D$ is the following:

$$\min_G \max_D V(G, D) := \mathbb{E}_{\mathbf{x} \sim p_{\text{data}}}[\log(D(\mathbf{x}))] + \mathbb{E}_{\mathbf{z} \sim p(\mathbf{z})}[\log(1 - D(G(\mathbf{z})))]. \tag{1}$$

For a fixed $G$, the inner maximization objective (1) with respect to $D$ leads to the following *optimal discriminator* $D_G^*$, and consequently the outer minimization objective with respect to $G$ becomes to minimize the *Jensen-Shannon divergence* between $p_{\text{data}}$ and $p_g$: $D_G^* := \max_D V(G, D) = \frac{p_{\text{data}}}{p_{\text{data}} + p_g}$. Although this formulation (1) theoretically guarantees $p_g^* = p_{\text{data}}$ as the global optimum, the *non-saturating loss* (Goodfellow et al., 2014) is more favored in practice for better optimization stability:

$$\max_D L(D) := V(G, D), \text{ and } \min_G L(G) := -\mathbb{E}_{\mathbf{z}}[\log(D(G(\mathbf{z})))]. \tag{2}$$

Here, compared to (1), $G$ is now optimized to let $D$ to classify $G(\mathbf{z})$ as 1, *i.e.*, the "real".

**Contrastive representation learning.** Consider two random variables $\mathbf{v}^{(1)}$ and $\mathbf{v}^{(2)}$, which are often referred as *views*. Generally speaking, *contrastive learning* aims to extract a useful representation of $\mathbf{v}^{(1)}$ and $\mathbf{v}^{(2)}$ from learning a function that identifies whether a given sample is from

$p(\mathbf{v}^{(1)})p(\mathbf{v}^{(2)}|\mathbf{v}^{(1)})$ or $p(\mathbf{v}^{(1)})p(\mathbf{v}^{(2)})$, *i.e.*, whether two views are dependent or not. More specifically, the function estimates the *mutual information* $I(\mathbf{v}^{(1)};\mathbf{v}^{(2)})$ between the two views. To this end, Oord et al. (2018) proposed to minimize *InfoNCE* loss, which turns out to maximize a lower bound of $I(\mathbf{v}^{(1)};\mathbf{v}^{(2)})$. Formally, for a given $\boldsymbol{v}_i^{(1)} \sim p(\mathbf{v}^{(1)})$ and $\boldsymbol{v}_i^{(2)} \sim p(\mathbf{v}^{(2)}|\boldsymbol{v}_i^{(1)})$ while assuming $\boldsymbol{v}_j^{(2)} \sim p(\mathbf{v}^{(2)})$ for $j = 1, \cdots, K$, the InfoNCE loss is defined by:

$$L_{\text{NCE}}(\boldsymbol{v}_i^{(1)}; \boldsymbol{v}^{(2)}, s) := -\log \frac{\exp(s(\boldsymbol{v}_i^{(1)}, \boldsymbol{v}_i^{(2)}))}{\sum_{j=1}^{K} \exp(s(\boldsymbol{v}_i^{(1)}, \boldsymbol{v}_j^{(2)}))}, \tag{3}$$

where $s(\cdot, \cdot)$ is the score function that models the log-density ratio of $p(\mathbf{v}^{(2)}|\mathbf{v}^{(1)})$ to $p(\mathbf{v}^{(2)})$, possibly including some parametrized encoders for $\mathbf{v}^{(1)}$ and $\mathbf{v}^{(2)}$.

Many of recent unsupervised representation learning methods are based on this general framework of contrastive learning (Wu et al., 2018; Bachman et al., 2019; Hénaff et al., 2020; He et al., 2020; Chen et al., 2020a). In this paper, we focus on the one called *SimCLR* (Chen et al., 2020a), that adopts a wide range of independent data augmentations to define views: specifically, for a given set of data samples $\mathbf{x} = [\mathbf{x}_i]_{i=1}^{N}$, SimCLR applys two independent augmentations, namely $\mathbf{t}_1$ and $\mathbf{t}_2$, to the given data to obtain $\mathbf{v}^{(1)}$ and $\mathbf{v}^{(2)}$, *i.e.*, $(\mathbf{v}^{(1)}, \mathbf{v}^{(2)}) := (\mathbf{t}_1(\mathbf{x}), \mathbf{t}_2(\mathbf{x}))$. The actual loss of SimCLR is slightly different to InfoNCE, mainly due to practical considerations for sample efficiency:

$$L_{\text{SimCLR}}(\boldsymbol{v}^{(1)}, \boldsymbol{v}^{(2)}) := \frac{1}{2N} \sum_{i=1}^{N} \left( L_{\text{NCE}}(\boldsymbol{v}_i^{(1)}; [\boldsymbol{v}^{(2)}; \boldsymbol{v}_{-i}^{(1)}], s_{\text{SimCLR}}) + L_{\text{NCE}}(\boldsymbol{v}_i^{(2)}; [\boldsymbol{v}^{(1)}; \boldsymbol{v}_{-i}^{(2)}], s_{\text{SimCLR}}) \right), \tag{4}$$

where $\boldsymbol{v}_{-i} := \boldsymbol{v} \setminus \{\boldsymbol{v}_i\}$. For $s_{\text{SimCLR}}$, SimCLR specifies to use (a) an *encoder* network $f : \mathcal{X} \to \mathbb{R}^{d_e}$, (b) a small neural network called *projection head* $h : \mathbb{R}^{d_e} \to \mathbb{R}^{d_p}$, and (c) the *normalized temperature-scaled cross entropy* (NT-Xent). Putting altogether, $s_{\text{SimCLR}}$ is defined by:

$$s_{\text{SimCLR}}(\mathbf{v}^{(1)}, \mathbf{v}^{(2)}; f, h) := \frac{h(f(\mathbf{v}^{(1)})) \cdot h(f(\mathbf{v}^{(2)}))}{\tau \cdot ||h(f(\mathbf{v}^{(1)}))||_2 ||h(f(\mathbf{v}^{(2)}))||_2}, \tag{5}$$

where $\tau$ is a temperature hyperparameter. Once the training is done with respect to $L_{\text{SimCLR}}$, the projection head $h$ is discarded and $f$ is served as the learned representation for downstream tasks.

## 3 CONTRAD: CONTRASTIVE DISCRIMINATOR FOR GANS

We aim to develop a training scheme for GANs that is capable to extract more useful information under stronger data augmentation beyond the existing yet limited practices, *e.g.*, random translations up to few pixels (Zhang et al., 2020; Zhao et al., 2020c;a). For examples, one may consider to apply augmentations introduced in SimCLR (Chen et al., 2020a) for training GANs, *i.e.*, a stochastic composition of random resizing, cropping, horizontal flipping, color distortion, and Gaussian blurring. Indeed, existing approaches that handle data augmentation in GANs are not guranteed to work best in this harsh case (as also observed empirically in Table 5), and some of them are even expected to harm the performance: *e.g.*, applying consistency regularization (Zhang et al., 2020; Zhao et al., 2020c), which regularizes a discriminator to be *invariant* to these augmentations, could overly restrict the expressivity of the discriminator to learn meaningful features.

Our proposed architecture of *Contrastive Discriminator* (ContraD) is designed to alleviate such potential risks from using stronger data augmentations by incorporating the contrastive learning scheme of SimCLR (Chen et al., 2020a), which arguably works with those augmentations, and its design practices into the discriminator. More concretely, the main objective of ContraD is not to minimize the discriminator loss of GAN, but to learn a contrastive representation that is *compatible* to GAN. This means that the objective does not break the contrastive learning, while the representation still contains sufficient information to discriminate real and fake samples, so that a small neural network discriminator is enough to perform its task upon the representation. We describe how to learn such a representation of ContraD in Section 3.1, and how to incorporate this in the actual GAN training in Section 3.2. Finally, Section 3.3 introduces a natural application of ContraD for deriving conditional generative models from an unconditionally trained GAN. Figure 1 illustrates ContraD.

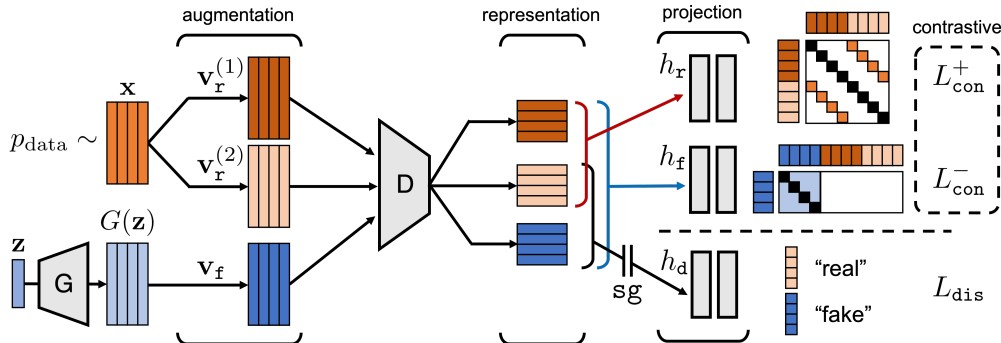

Figure 1: An overview of *Contrastive Discriminator* (ContraD). Overall, the representation of ContraD is not learned from the discriminator loss ($L_{\mathtt{dis}}$), but from two contrastive losses $L_{\mathtt{con}}^{+}$ and $L_{\mathtt{con}}^{-}$, each is for the real and fake samples, respectively. Here, $\mathtt{sg}(\cdot)$ denotes the stop-gradient operation.

### 3.1 CONTRASTIVE REPRESENTATION FOR GANS

Recall the standard training of GANs, *i.e.*, we train $G : \mathcal{Z} \rightarrow \mathcal{X}$ and $D : \mathcal{X} \rightarrow [0, 1]$ via optimizing (2) from a given dataset $\{\mathbf{x}_i\}_{i=1}^N$, and let $\mathcal{T}$ be a family of possible transforms for data augmentation, which we assume in this paper to be those used in SimCLR (Chen et al., 2020a). In order to define the training objective of ContraD, we start by re-defining the scalar-valued $D$ to have vector-valued outputs, *i.e.*, $D : \mathcal{X} \rightarrow \mathbb{R}^{d_e}$, to represent an encoder network of contrastive learning. Overall, the encoder network $D$ of ContraD is trained by minimizing two different *contrastive* losses: (a) the SimCLR loss (4) on the real samples, and (b) the supervised contrastive loss (Khosla et al., 2020) on the fake samples, which will be explained one-by-one in what follows.

**Loss for real samples.** By default, ContraD is built upon the SimCLR training: in fact, the training becomes equivalent to SimCLR if the loss (b) is missing, *i.e.*, without considering the fake samples. Here, we attempt to simply follow the contrastive training scheme for real samples $\boldsymbol{x}$, to keep open the possibility to improve the method by adopting other self-supervised learning methods with data augmentations (Chen et al., 2020b; Grill et al., 2020). More concretely, we first compute two independent views $\boldsymbol{v}_{\mathrm{r}}^{(1)}, \boldsymbol{v}_{\mathrm{r}}^{(2)} := \boldsymbol{t}_1(\mathbf{x}), \boldsymbol{t}_2(\mathbf{x})$ for real samples using $\boldsymbol{t}_1, \boldsymbol{t}_2 \sim \mathcal{T}$, and minimize:

$$L_{\mathtt{con}}^{+}(D, h_{\mathrm{r}}) := L_{\mathtt{SimCLR}}(\boldsymbol{v}_{\mathrm{r}}^{(1)}, \boldsymbol{v}_{\mathrm{r}}^{(2)}; D, h_{\mathrm{r}}), \tag{6}$$

where $h_{\mathrm{r}} : \mathbb{R}^{d_e} \rightarrow \mathbb{R}^{d_p}$ is a projection head for this loss.

**Loss for fake samples.** Although the representation learned via $L_{\mathtt{con}}^{+}$ (6) may be enough, *e.g.*, for discriminating two independent real samples, it does not give any guarantee that this representation would still work for discriminating real and fake samples, which is needed as a GAN discriminator. We compensate this by considering an auxiliary loss $L_{\mathtt{con}}^{-}$ that regularizes the encoder to keep necessary information to discriminate real and fake samples. Here, several design choices can be possible: nevertheless, we observe that one should choose the loss deliberately, otherwise the contrastive representation from $L_{\mathtt{con}}^{+}$ could be significantly affected. For example, we found using the original GAN loss (2) for $L_{\mathtt{con}}^{-}$ could completely negate the effectiveness of ContraD.

In this respect, we propose to use the *supervised contrastive loss* (Khosla et al., 2020) over fake samples to this end, an extended version of contrastive loss to support supervised learning by allowing more than one view to be positive, so that views of the same label can be attracted to each other in the embedding space. In our case, we assume all the views from fake samples have the same label against those from real samples. Formally, for each $\boldsymbol{v}_i^{(1)}$, let $V_{i+}^{(2)}$ be a subset of $\boldsymbol{v}^{(2)}$ that represent the positive views for $\boldsymbol{v}_i^{(1)}$. Then, the supervised contrastive loss is defined by:

$$L_{\mathtt{SupCon}}(\boldsymbol{v}_i^{(1)}, \boldsymbol{v}^{(2)}, V_{i+}^{(2)}) := -\frac{1}{|V_{i+}^{(2)}|} \sum_{\boldsymbol{v}_{i+}^{(2)} \in V_{i+}^{(2)}} \log \frac{\exp(s_{\mathtt{SimCLR}}(\boldsymbol{v}_i^{(1)}, \boldsymbol{v}_{i+}^{(2)}))}{\sum_j \exp(s_{\mathtt{SimCLR}}(\boldsymbol{v}_i^{(1)}, \boldsymbol{v}_j^{(2)}))}. \tag{7}$$

Using the notation, we define the ContraD loss for fake samples as follows:

$$L_{\mathrm{con}}^-(D, h_{\mathrm{f}}) := \frac{1}{N} \sum_{i=1}^N L_{\mathtt{SupCon}}(\boldsymbol{v}_{\mathrm{f},i}, [\boldsymbol{v}_{\mathrm{f},-i}; \boldsymbol{v}_{\mathrm{r}}^{(1)}; \boldsymbol{v}_{\mathrm{r}}^{(2)}], [\boldsymbol{v}_{\mathrm{f},-i}]; D, h_{\mathrm{f}}), \tag{8}$$

where again $\boldsymbol{v}_{\mathrm{f}} := \boldsymbol{t}_3(G(\mathbf{z}))$ is a random view of fake samples, and $\boldsymbol{v}_{-i} := \boldsymbol{v} \setminus \{\boldsymbol{v}_i\}$. Remark that we use an independent projection header $h_{\mathrm{f}} : \mathbb{R}^{d_e} \to \mathbb{R}^{d_p}$ instead of $h_{\mathrm{r}}$ in (6) for this loss.

There can be other variants for $L_{\mathrm{con}}^-$, as long as (a) it leads $D$ to discriminate real and fake samples, while (b) not compromising the representation from $L_{\mathrm{con}}^+$. For example, one can incorporate another view of fake samples to additionally perform SimCLR similarly to $L_{\mathrm{con}}^+$ (6). Nevertheless, we found the current design (8) is favorable among such variants considered, both in terms of performance under practice of GANs, and computational efficiency from using only a single view.

To sum up, ContraD learns its contrastive representation by minimizing the following loss:

$$L_{\mathrm{con}}(D, h_{\mathrm{r}}, h_{\mathrm{f}}) := L_{\mathrm{con}}^+(D, h_{\mathrm{r}}) + \lambda_{\mathrm{con}} L_{\mathrm{con}}^-(D, h_{\mathrm{f}}), \tag{9}$$

where $\lambda_{\mathrm{con}} > 0$ is a hyperparameter. Nevertheless, we simply use $\lambda_{\mathrm{con}} = 1$ in our experiments.

## 3.2 TRAINING GANs WITH CONTRAD

The contrastive loss defined in (9) is jointly trained under the standard framework of GAN, *e.g.*, we present in this paper the case of non-saturating loss (2), for training the generator network $G$. To obtain a (scalar) discriminator score needed to define the GAN loss, we simply use an additional *discriminator head* $h_{\mathrm{d}} : \mathbb{R}^{d_e} \to \mathbb{R}$ upon the contrastive representation of $D$. Here, a key difference of ContraD training from other GANs is that ContraD only optimizes the parameters of $h_{\mathrm{d}}$ for minimizing the GAN loss: in practical situations of using stochastic gradient descent for the training, this can be implemented by *stopping gradient* before feeding inputs to $h_{\mathrm{d}}$. Therefore, the discriminator loss of ContraD is defined as follows:

$$L_{\mathrm{dis}}(h_{\mathrm{d}}) := -\mathbb{E}_{\mathbf{r}_{\mathrm{r}}}[\log(\sigma(h_{\mathrm{d}}(\mathbf{r}_{\mathrm{r}})))] - \mathbb{E}_{\mathbf{r}_{\mathrm{f}}}[\log(1 - \sigma(h_{\mathrm{d}}(\mathbf{r}_{\mathrm{f}})))], \tag{10}$$

where $\mathbf{r}_{\mathrm{r}} = \mathtt{sg}(D(\mathbf{v}_{\mathrm{r}}))$ and $\mathbf{r}_{\mathrm{f}} = \mathtt{sg}(D(\mathbf{v}_{\mathrm{f}}))$. Here, $\sigma(\cdot)$ denotes the sigmoid function, $\mathbf{v}_{\mathrm{r}}$ and $\mathbf{v}_{\mathrm{f}}$ are random views of real and fake samples augmented via $\mathcal{T}$, respectively, and $\mathtt{sg}(\cdot)$ is the stop-gradient operation. Combined with the contrastive training loss (9), the joint training loss $L_D$ for a single discriminator update is the following:

$$L_D := L_{\mathrm{con}} + \lambda_{\mathrm{dis}} L_{\mathrm{dis}} = L_{\mathrm{con}}^+ + \lambda_{\mathrm{con}} L_{\mathrm{con}}^- + \lambda_{\mathrm{dis}} L_{\mathrm{dis}}, \tag{11}$$

where $\lambda_{\mathrm{dis}} > 0$ is a hyperparameter. Again, as like $\lambda_{\mathrm{con}}$, we simply use $\lambda_{\mathrm{dis}} = 1$ in our experiments.

Finally, the loss for the generator $G$ can be defined simply like other standard GANs using the discriminator score, except that we also augment the generated samples $G(\mathbf{z})$ to $\mathbf{v}_{\mathrm{f}}$, in a similar way to DiffAug (Zhao et al., 2020a) or ADA (Karras et al., 2020a): namely, in case of the non-saturating loss (Goodfellow et al., 2014), we have:

$$L_G := -\mathbb{E}_{\mathbf{v}_{\mathrm{f}}}[\log(\sigma(h_{\mathrm{d}}(D(\mathbf{v}_{\mathrm{f}}))))]. \tag{12}$$

Note that the form (12) is equivalent to the original non-saturating loss (2) if we regard $\sigma(h_{\mathrm{d}}(D(\cdot))) : \mathcal{X} \to [0, 1]$ as a single discriminator. Algorithm 1 in Appendix A describes a concrete training procedure of GANs with ContraD using Adam optimizer (Kingma & Ba, 2014).

## 3.3 SELF-CONDITIONAL SAMPLING WITH CONTRAD

Apart from the improved training of GANs, the learned contrastive representation of ContraD could offer some additional benefits in practical scenarios compared to the standard GANs. In this section, we present an example of how one could further utilize this additional information to derive a conditional generative model from an unconditionally-trained GAN with ContraD. More specifically, here we consider a variant of *discriminator-driven latent sampling* (DDLS) (Che et al., 2020) to incorporate a given representation vector from the ContraD encoder as a conditional information. Originally, DDLS attempts to improve the sample quality of a pre-trained GAN via a Langevin sampling (Welling & Teh, 2011) on the latent space of the following form:

$$\mathbf{z}_{t+1} := \mathbf{z}_t - \frac{\varepsilon}{2} \nabla_{\mathbf{z}_t} E(\mathbf{z}_t) + \sqrt{\varepsilon} \mathbf{n}_t, \mathbf{n}_t \sim \mathcal{N}(0, 1), \tag{13}$$

Table 1: Comparison of the best FID score and IS on unconditional image generation of CIFAR-10 and CIFAR-100. Values in the rows marked by * are from those reported in its reference.

| Architecture | Method | Augment. | CIFAR-10 | | CIFAR-100 | |
|---|---|---|---|---|---|---|
| | | | FID $\downarrow$ | IS $\uparrow$ | FID $\downarrow$ | IS $\uparrow$ |
| $G$: SNDCGAN $D$: SNDCGAN | - | - | 26.6 | 7.38 | 28.5 | 7.25 |
| | CR (Zhang et al., 2020) | HFlip, Trans | 19.5 | 7.87 | 22.2 | 7.91 |
| | bCR (Zhao et al., 2020c) | HFlip, Trans | 14.0 | 8.35 | 19.2 | 8.46 |
| | DiffAug (Zhao et al., 2020a) | Trans, CutOut | 22.9 | 7.64 | 27.0 | 7.47 |
| | **ContraD (ours)** | **SimCLR** | **10.9** | **8.78** | **15.2** | **9.09** |
| $G$: SNDCGAN $D$: SNResNet-18 | - | - | 41.3 | 6.33 | 52.3 | 5.24 |
| | CR (Zhang et al., 2020) | HFlip, Trans | 32.1 | 7.08 | 36.5 | 6.55 |
| | bCR (Zhao et al., 2020c) | HFlip, Trans | 22.8 | 7.29 | 28.2 | 7.30 |
| | DiffAug (Zhao et al., 2020a) | Trans, CutOut | 59.5 | 5.62 | 58.7 | 5.39 |
| | **ContraD (ours)** | **SimCLR** | **9.86** | **9.09** | **15.0** | **9.56** |
| $G$: StyleGAN2 $D$: StyleGAN2 | - | - | 11.1 | 9.18 | 16.5 | 9.51 |
| | DiffAug* (Zhao et al., 2020a) | Trans, CutOut | 9.89 | 9.40 | 15.2 | **10.0** |
| | **ContraD (ours)** | **SimCLR** | **9.80** | **9.47** | **14.1** | **10.0** |

Table 2: Comparison of the best FID score and IS on unconditional image generation of CelebA-HQ-128 with SNDCGAN. We report the mean and standard deviation of best scores across 3 trials.

| CelebA-HQ | W/O | Hinge | CR | bCR | ContraD (ours) |
|---|---|---|---|---|---|
| FID $\downarrow$ | $24.8_{\pm1.69}$ | $26.4_{\pm1.32}$ | $20.9_{\pm0.37}$ | $19.5_{\pm0.08}$ | $\mathbf{17.1}_{\pm1.38}$ |
| IS $\uparrow$ | $2.06_{\pm0.06}$ | $2.05_{\pm0.06}$ | $2.11_{\pm0.09}$ | $2.21_{\pm0.10}$ | $\mathbf{2.44}_{\pm0.06}$ |

where $E(\mathbf{z}) := -\log p(\mathbf{z}) - d(G(\mathbf{z}))$ and $d(\cdot)$ denotes the logit of $D$. In this manner, for our ContraD, we propose to use the following "conditional" version of energy function $E(\mathbf{z}, v)$ for an arbitrary vector $v \in \mathbb{R}^{d_e}$ in the ContraD representation space, called *conditional DDLS* (cDDLS):

$$E(\mathbf{z}, v) := -\log p(\mathbf{z}) - h_{\mathsf{d}}(D(G(\mathbf{z}))) - \lambda(v \cdot D(G(\mathbf{z}))). \tag{14}$$

In our experiments, for instance but not limited to, we show that using the learned weights obtained from linear evaluation as conditional vectors successfully recovers class-conditional generations from an unconditional ContraD model.

## 4 EXPERIMENTS

We verify the effectiveness of our ContraD in three different aspects: (a) its performance on image generation compared to other techniques for data augmentations in GANs, (b) the quality of representations learned from ContraD in terms of linear evaluation, and (c) its ability of self-conditional sampling via cDDLS. Overall, we constantly observe integrating contrastive learning into GANs and *vice versa* have positive effects to each other: (a) for image generation, ContraD enables a simple SNDCGAN to outperform a state-of-the-art architecture of StyleGAN2 on CIFAR-10/100; (b) in case of linear evaluation, on the other hand, we observe the representation from ContraD could outperform those learned from SimCLR. Finally, we perform an ablation study to further understand each of the components we propose. We provide the detailed specification on the experimental setups, *e.g.*, architectures, training configurations and hyperparameters in Appendix F.

We consider a variety of datasets including CIFAR-10/100 (Krizhevsky, 2009), CelebA-HQ-128 (Lee et al., 2020), AFHQ (Choi et al., 2020) and ImageNet (Russakovsky et al., 2015) in our experiments, mainly with three well-known GAN architectures: SNDCGAN (Miyato et al., 2018), StyleGAN2 (Karras et al., 2020b) and BigGAN (Brock et al., 2019).[1] We measure *Fréchet Inception distance* (FID) (Heusel et al., 2017) and *Inception score* (IS) (Salimans et al., 2016) as quantitative metrics to evaluate generation quality.[2] We compute FIDs between the *test set* and generated samples of the same size. We follow the best hyperparameter practices explored by Kurach et al. (2019) for SNDCGAN models on CIFAR and CelebA-HQ-128 datasets. For StyleGAN2, on the

---

[1]Results on AFHQ and ImageNet can be found in Appendix B and C, respectively.

[2]We use the official InceptionV3 model in TensorFlow to compute both FID and IS in our experiments.

Table 3: Comparison of classification accuracy under linear evaluation protocol on CIFAR-10 and CIFAR-100. We report the mean and standard deviation across 3 runs of the evaluation.

| Dataset | Training | SNDCGAN | SNResNet-18 | StyleGAN2 |
|---------|----------|---------|-------------|-----------|
| CIFAR-10 | SimCLR ($\lambda_{\texttt{con}} = \lambda_{\texttt{dis}} = 0$) | $72.9_{\pm 0.02}$ | $80.3_{\pm 0.05}$ | $86.2_{\pm 0.06}$ |
|  | **ContraD (ours)** | $\mathbf{77.5}_{\pm 0.20}$ | $\mathbf{85.7}_{\pm 0.10}$ | $\mathbf{88.6}_{\pm 0.06}$ |
| CIFAR-100 | SimCLR ($\lambda_{\texttt{con}} = \lambda_{\texttt{dis}} = 0$) | $30.8_{\pm 0.11}$ | $41.2_{\pm 0.06}$ | $61.1_{\pm 0.06}$ |
|  | **ContraD (ours)** | $\mathbf{37.4}_{\pm 0.06}$ | $\mathbf{51.1}_{\pm 0.18}$ | $\mathbf{68.1}_{\pm 0.07}$ |

other hand, we follow the training details of Zhao et al. (2020a) in their CIFAR experiments. All the training details are shared across methods per experiment, but the choice of batch size for training ContraD: we generally observe that ContraD greatly benefits from using larger batch (Appendix E), similarly to SimCLR (Chen et al., 2020a), but in contrast to the common practice of GAN training. Therefore, in our experiments, we consider to use $2\times$ to $8\times$ larger batch sizes for training ContraD, where the details are specified in Appendix F.

## 4.1 RESULTS

**Image generation on CIFAR-10/100.** CIFAR-10 and CIFAR-100 (Krizhevsky, 2009) consist of 60K images of size $32 \times 32$ in 10 and 100 classes, respectively, 50K for training and 10K for testing. We start by comparing ContraD on CIFAR-10/100 with three recent methods that applies data augmentation in GANs: (a) Consistency Regularization (CR) (Zhang et al., 2020), (b) the "balanced" Consistency Regularization (bCR) (Zhao et al., 2020c), and (c) Differentiable Augmentation (DiffAug) (Zhao et al., 2020a) as baselines.[3] We follow their best pratices on which data augmentations to use: specifically, we use a combination of horizontal flipping (HFlip) and random translation up to 4 pixels (Trans) for CR and bCR, and Trans + CutOut (DeVries & Taylor, 2017) for DiffAug for CIFAR datasets. We mainly consider SNDCGAN and StyleGAN2 for training, but in addition, we further consider a scenario that the discriminator architecture of SNDCGAN is scaled up into *SNResNet-18*, which is identical to ResNet-18 (He et al., 2016) without batch normalization (Ioffe & Szegedy, 2015) but with spectral normalization (Miyato et al., 2018), in order to see that how each training method behaves when the capacity of generator and discriminator are significantly different.

Table 1 shows the results: overall, we observe our ContraD uniformly improves the baseline GAN training by a large margin, significantly outperforming other baselines as well. Indeed, our results on SNDCGAN already achieve better FID than those from the baseline training of StyleGAN2. Moreover, when the discriminator of SNDCGAN is increased to SNResNet-18, ContraD could further improve its results, without even modifying any of the hyperparameters. It is particularly remarkable that all the other baselines could not improve their results in this setup: A further ablation study presented in Section 4.2 reveals that *stopping gradients* before the discriminator head in our design is an important factor to benefit from SNResNet-18. Finally, our ContraD can also be applied to a larger, sophisticated StyleGAN2, further improving both FID and IS compared to DiffAug.

**Image generation on CelebA-HQ-128.** We also perform experiments on CelebA-HQ-128 (Lee et al., 2020), which consist of 30K human faces of size $128 \times 128$, to see that our design of ContraD still applies well for high-resolution datasets. We split 3K images out of CelebA-HQ-128 for testing, and use the remaining for training. We use SNDCGAN for this experiment, and compare FID and IS to (a) the default non-saturating loss (W/O) (Goodfellow et al., 2014), (b) the hinge loss (Hinge) (Lim & Ye, 2017; Tran et al., 2017), (c) CR and (d) bCR. Similarly to the CIFAR experiments, we use the "HFlip+Trans" augmentation for both CR and bCR, but for CelebA-HQ-128 we allow them to translate up to 16 pixels, following (Zhang et al., 2020). Again, as shown in Table 2, we confirm that ContraD could still improve training of GANs compared to other methods, even with this high-resolution, yet with limited samples and a straightforward choice of architecture.

**Linear evaluation.** Recall that the training objective of ContraD we propose in (11) can be simply reduced into the SimCLR loss (Chen et al., 2020a) without considering the fake samples, *i.e.*, by letting $\lambda_{\texttt{con}} = \lambda_{\texttt{dis}} = 0$. A natural question from here is whether those fake samples could help to

---

[3]Here, we notice that ADA (Karras et al., 2020a), a concurrent work to DiffAug, also offers a highly similar method to DiffAug, but with an additional heuristic of tuning augmentations on-the-fly with a validation dataset. We consider this addition is orthogonal to ours, and focus on comparing DiffAug for a clearer comparison in the CIFAR experiments. A more direct comparison with ADA, nevertheless, can be found in Appendix B.

Table 4: Comparison of FID (lower is better) from the class-wise subsets of CIFAR-10 test set (1K images per class). "Random" indicates unconditional generation, and "Train-1K" indicates a 1K random subsamples from the CIFAR-10 train set of the same class.

| $D$ Arch. | Sampling | Plane | Car | Bird | Cat | Deer | Dog | Frog | Horse | Ship | Truck | Mean |
|---|---|---|---|---|---|---|---|---|---|---|---|---|
| SNDCGAN | Random | 110.6 | 125.9 | 103.8 | 104.4 | 107.0 | 121.0 | 134.1 | 120.1 | 128.8 | 130.0 | 118.6 |
| | + cDDLS | **65.3** | **72.1** | **63.9** | **69.4** | **57.5** | **69.0** | **107.0** | **61.5** | **58.2** | **53.6** | **67.8** |
| SNResNet-18 | Random | 117.6 | 136.3 | 100.5 | 99.6 | 98.6 | 115.0 | 128.2 | 111.7 | 140.9 | 140.3 | 118.9 |
| | + cDDLS | **67.1** | **65.8** | **59.8** | **63.8** | **59.7** | **61.9** | **68.8** | **59.9** | **58.9** | **51.6** | **61.7** |
| - | Train-1K | 38.9 | 28.4 | 41.4 | 49.7 | 36.4 | 40.2 | 41.7 | 33.6 | 30.7 | 24.6 | 36.6 |

Table 5: Comparison of the best FID and IS on CIFAR-10 with SND-CGAN when stronger (bCR and DiffAug) or weaker augmentations (ContraD) are used for each method.

| Method | Augment. | FID ↓ | IS ↑ |
|---|---|---|---|
| bCR | HFlip, Trans | 14.0 | 8.35 |
| + Aug. | SimCLR | 20.6 | 7.44 |
| DiffAug | Trans, CutOut | 22.9 | 7.64 |
| + Aug. | SimCLR | 21.9 | 7.42 |
| **ContraD** | **SimCLR** | **10.9** | **8.78** |
| – Aug. | HFlip, Trans | 13.7 | 8.54 |

Table 6: Comparison of the best FID score and IS on CIFAR-10 for ablations of our proposed components. All the models are trained with batch size 256. For the ablation of "MLP $h_d$", we replace $h_d$ with a linear model. "$\mathrm{sg}(\cdot)$" indicates the use of stop-gradient operation.

| | | | | $D$: SNDCGAN | | $D$: SNResNet-18 | |
|---|---|---|---|---|---|---|---|
| MLP $h_d$ | $\mathrm{sg}(\cdot)$ | $L_{con}^+$ | $L_{con}^-$ | FID ↓ | IS ↑ | FID ↓ | IS ↑ |
| ✓ | ✓ | ✓ | ✓ | **11.1** | **8.62** | **10.6** | **8.99** |
| ✗ | ✓ | ✓ | ✓ | 185 | 3.43 | 274 | 2.09 |
| ✓ | ✗ | ✓ | ✓ | 11.6 | 8.61 | 28.0 | 7.52 |
| ✓ | ✓ | ✗ | ✓ | 11.9 | 8.45 | 182 | 2.02 |
| ✓ | ✓ | ✓ | ✗ | 210 | 1.93 | 232 | 2.56 |

improve the representation of SimCLR: we verify this by comparing the *linear evaluation* performance (Kolesnikov et al., 2019; Chen et al., 2020a; Grill et al., 2020) of the learned representation of ContraD to those from SimCLR: specifically, linear evaluation measures the test accuracy that a linear classifier on the top of given (frozen) representation could maximally achieve for a given (downstream) labeled data. Following this protocol, we compare the linear evaluation of a ContraD model with its ablation that $\lambda_{con}$ and $\lambda_{dis}$ are set to 0 during its training. We consider three discriminator architectures and two datasets for this experiments: namely, we train SNDCGAN, SNResNet-18, and StyleGAN2 models on CIFAR-10 and CIFAR-100. In this experiment, we use batch size 256 for training all the SNDCGAN and SNResNet-18 models, while in case of Style-GAN2 we use batch size 64 instead. Table 3 summarizes the results, and somewhat interestingly, it shows that ContraD significantly improves the SimCLR counterpart in terms of linear evaluation for all the models tested: in other words, we observe that keeping discriminative features between real and fake samples in their representation, which is what ContraD does in addition to SimCLR, can be beneficial to improve the contrastive learning itself. Again, our observation supports a great potential of ContraD not only in the context of GANs, but also in the contrastive learning.

**Conditional DDLS with ContraD.** We further evaluate the performance of conditional DDLS (cD-DLS) (14) proposed for ContraD, with a task of deriving class-conditional generative models for a given (unconditionally-trained) GAN model. Here, we measure the performance of this conditional generation by computing the *class-wise* FIDs to the test set, *i.e.*, FIDs computed for each subsets of the same class in the test set. We test cDDLS on SNDCGAN and SNResNet-18 ContraD models trained on CIFAR-10. To perform a conditional generation for a specific class, we simply take a learned linear weight obtained from the linear evaluation protocol for that class. The results summarized in Table 4 show that applying cDDLS upon unconditional generation significantly improves class-wise FIDs for all the classes in CIFAR-10. Some of the actual samples conditionally generated from an SNResNet-18 ContraD model can be found in Figure 5 of Appendix D.

## 4.2 ABLATION STUDY

We conduct an ablation study for a more detailed analysis on the proposed method. For this section, unless otherwise specified, we perform experiments on CIFAR-10 with SNDCGAN architecture.

**Stronger augmentations for other methods.** One of the key characteristics of ContraD compared to the prior works is the use of stronger data augmentations for training GANs, *e.g.*, we use the augmentation pipeline of SimCLR (Chen et al., 2020a) in our experiments. In Table 5, we examine the performance bCR (Zhao et al., 2020c) and DiffAug (Zhao et al., 2020a) when this stronger

augmentation is applied, and shows that the SimCLR augmentation could not meaningfully improve FID or IS for both methods. Unlike these methods, our ContraD effectively handles the SimCLR augmentations without overly regularizing the discriminator, possibly leveraging many components of contrastive learning: *e.g.*, the normalized loss, or the use of separate projection heads.

**Weaker augmentations for ContraD.** On the other side, we also consider a case when ContraD is rather trained with a weaker data augmentation, here we consider "HFlip, Trans" instead of "Sim-CLR", in Table 5: even in this case, we observe ContraD is still as good as (or better in terms of IS) a strong baseline of bCR. The degradation in FID (and IS) compared to "ContraD + SimCLR" is possibly due to that SimCLR requires strong augmentations to learn a good representation (Chen et al., 2020a), *e.g.*, we also observe there is a degradation in the linear evaluation performances, $77.5 \rightarrow 72.9$, when "HFlip, Trans" is used.

**MLP discriminator head.** In our experiments, we use a 2-layer network for the discriminator head $h_\mathtt{d}$. As shown in Table 6, this choice can be minimal: replacing $h_\mathtt{d}$ with a linear model severely breaks the training. Conversely, although not presented here, we have also observed that MLPs of more than two layers neither give significant gains on the performance in our experimental setup.

**Stopping gradients.** We use the stop-gradient operation $\mathtt{sg}(\cdot)$ before the discriminator head $h_\mathtt{d}$ in attempt to decouple the effect of GAN loss to the ContraD representation. Table 6 also reports the ablation of this component: overall, not using $\mathtt{sg}(\cdot)$ does not completely break the training, but it does degrade the performance especially when a deeper discriminator is used, *e.g.*, SNResNet-18.

**Contrastive losses.** Recall that the representation of ContraD is trained by two losses, namely $L_\mathtt{con}^+$ and $L_\mathtt{con}^-$ (Section 3.1). From an ablation study on each loss, as presented in Table 6, we observe that both of the losses are indispensable for our training to work in a larger discriminator architecture.

**Separate projection headers.** As mentioned in Section 3.1, we use two independent projection headers, namely $h_\mathtt{r}$ and $h_\mathtt{f}$, to define $L_\mathtt{con}^+$ and $L_\mathtt{con}^-$, respectively. We have observed this choice is empirically more stable than sharing them: *e.g.*, under $h_\mathtt{r} = h_\mathtt{f}$, we could not obtain a reasonable performance when only $D$ is increased to SNResNet-18 on CIFAR-10 with SNDCGAN. Intuitively, an optimal embedding (after projection) from $L_\mathtt{con}^-$ can be harmful to SimCLR, as it encourages the embedding of real samples to degenerate into a single point, *i.e.*, the opposite direction of SimCLR.

## 5 CONCLUSION AND FUTURE WORK

In this paper, we explore a novel combination of GAN and contrastive learning, two important, yet seemingly different paradigms in unsupervised representation learning. They both have been independently observed the crucial importance of maintaining consistency to data augmentations in their representations (Zhang et al., 2020; Tian et al., 2020), and here we further observe that these two representations complement each other when combined upon the shared principle of *view-invariant* representations: although we have put more efforts in this work to verify the effectiveness of contrastive learning on GANs, we do observe the opposite direction through our experiments, and scaling up our method to compete with state-of-the-art self-supervised learning benchmarks (Chen et al., 2020b; Grill et al., 2020; Caron et al., 2020) would be an important future work.

We believe our work also suggests several interesting ideas to explore in future research in both sides of GAN and contrastive learning: for example, our new design of introducing a small header to minimize the GAN loss upon other (*e.g.*, contrastive) representation is a promising yet unexplored way of designing a new GAN architecture. For the SimCLR side, on the other hand, we suggest a new idea of incorporating "fake" samples for contrastive learning, which is also an interesting direction along with other recent attempts to improve the efficiency of negative sampling in contrastive learning, *e.g.*, via hard negative mining (Kalantidis et al., 2020; Robinson et al., 2021).

ACKNOWLEDGMENTS

This work was supported by Samsung Advanced Institute of Technology (SAIT). This work was also partly supported by Institute of Information & Communications Technology Planning & Evaluation (IITP) grant funded by the Korea government (MSIT) (No.2019-0-00075, Artificial Intelligence Graduate School Program (KAIST)). The authors would like to thank Minkyu Kim for helping additional experiments in preparation of the camera-ready revision, and thank the anonymous reviewers for their valuable comments to improve our paper.

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

## A   TRAINING PROCEDURE OF CONTRAD

**Algorithm 1** GANs with Contrastive Discriminator (ContraD)

**Require:** generator $G$, discriminator $D$, Adam hyperparameters $\alpha, \beta_1, \beta_2$, number of $D$ updates
per $G$ update $N_D$, family of data augmentations $\mathcal{T}$, $\lambda > 0$.

1: **for** # training iterations **do**
2:     **for** $t = 1$ **to** $N_D$ **do**
3:         Sample $\boldsymbol{x} \sim p_{\text{data}}(\mathbf{x})$ and $\boldsymbol{z} \sim p(\mathbf{z})$
4:         Sample $\boldsymbol{t}_1, \boldsymbol{t}_2, \boldsymbol{t}_3 \sim \mathcal{T}$
5:         $\boldsymbol{v}_{\text{r}}^{(1)}, \boldsymbol{v}_{\text{r}}^{(2)}, \boldsymbol{v}_{\text{f}} \leftarrow \boldsymbol{t}_1(\boldsymbol{x}), \boldsymbol{t}_2(\boldsymbol{x}), \boldsymbol{t}_3(G(\boldsymbol{z}))$
6:         $L_{\text{con}}^{+} \leftarrow L_{\text{SimCLR}}(\boldsymbol{v}_{\text{r}}^{(1)}, \boldsymbol{v}_{\text{r}}^{(2)}; D, h_{\text{r}})$
7:         $L_{\text{con}}^{-} \leftarrow \frac{1}{N} \sum_{i=1}^{N} L_{\text{SupCon}}(\boldsymbol{v}_{\text{f},i}, [\boldsymbol{v}_{\text{f},-i}; \boldsymbol{v}_{\text{r}}^{(1)}; \boldsymbol{v}_{\text{r}}^{(2)}], [\boldsymbol{v}_{\text{f},-i}]; D, h_{\text{f}})$
8:         $\boldsymbol{r}_{\text{r}}, \boldsymbol{r}_{\text{f}} \leftarrow \texttt{sg}(D(\boldsymbol{v}_{\text{r}}^{(2)})), \texttt{sg}(D(\boldsymbol{v}_{\text{f}}))$
9:         $L_{\text{dis}} \leftarrow -\mathbb{E}_{\boldsymbol{r}_{\text{r}}}[\log(\sigma(h_{\text{d}}(\boldsymbol{r}_{\text{r}})))] - \mathbb{E}_{\boldsymbol{r}_{\text{f}}}[\log(1 - \sigma(h_{\text{d}}(\boldsymbol{r}_{\text{f}})))]$
10:         $L_D \leftarrow L_{\text{con}}^{+} + \lambda_{\text{con}} L_{\text{con}}^{-} + \lambda_{\text{dis}} L_{\text{dis}}$
11:         $D, h_{\text{r}}, h_{\text{f}}, h_{\text{d}} \leftarrow \text{Adam}([D, h_{\text{r}}, h_{\text{f}}, h_{\text{d}}], L_D; \alpha, \beta_1, \beta_2)$
12:     **end for**
13:     $L_G \leftarrow -\mathbb{E}_{\boldsymbol{v}_{\text{f}}}[\log(\sigma(h_{\text{d}}(D(\boldsymbol{v}_{\text{f}}))))]$
14:     $G \leftarrow \text{Adam}(G, L_G; \alpha, \beta_1, \beta_2)$
15: **end for**

## B   RESULTS ON AFHQ DATASETS

We evaluate our training method on the Animal Faces-HQ (AFHQ) dataset (Choi et al., 2020), which consists of ∼15,000 samples of animal-face images at 512×512 resolution, to further verify the effectiveness of ContraD on higher-resolution, yet limited-sized datasets. We partition the dataset by class labels into three sub-datasets, namely AFHQ-Dog (4,739 samples), AFHQ-Cat (5,153 samples) and AFHQ-Wild (4,738 samples), and evaluate FIDs on these datasets. In this experiment, we compare our method with ADA (Karras et al., 2020a), another recent data augmentation scheme for GANs (concurrently to DiffAug (Zhao et al., 2020a) considered in Section 4), which includes a dynamic adaptation of augmentation pipeline during training. We follow the training details of ADA particularly for this experiment: specifically, we train StyleGAN2 with batch size 64 for 25M training samples, using Adam with $(\alpha, \beta_1, \beta_2) = (0.0025, 0.0, 0.99)$, $R_1$ regularization with $\gamma = 0.5$, and exponential moving average on the generator weights with half-life of 20K samples. For ContraD training on AFHQ datasets, we consider to additionally use CutOut (DeVries & Taylor, 2017) of $p = 0.5$ upon the SimCLR augmentation, which we observe an additional gain in FID. We also follow the way of computing FIDs as per Karras et al. (2020a) here for a fair comparison, *i.e.*, by comparing 50K of generated samples and all the training samples (∼5K per dataset).

Table 7 compares our ContraD models with the results reported by Karras et al. (2020b) where the models are available at `https://github.com/NVlabs/stylegan2-ada`. Overall, we consistently observe that ContraD improves the baseline StyleGAN2, even achieving significantly better FIDs than ADA on AFHQ-Dog and AFHQ-Wild. These results confirms the effectiveness of ContraD on higher-resolution datasets, especially under the regime of limited data. Qualitative comparisons can be found in Figure 6, 7, and 8 of Appendix D.

Table 7: Comparison of the best FID score (lower is better) on unconditional image generation of AFHQ datasets (Choi et al., 2020) with StyleGAN2. Values in the rows marked by * are from those reported in its reference. We set our results bold-faced whenever the value improves the StyleGAN2 baseline ("Baseline"). Underlined indicates the best score among tested for each dataset.

| Architecture | Method | DOG | CAT | WILD |
|---|---|---|---|---|
| StyleGAN2 | Baseline* (Karras et al., 2020b) | 19.4 | 5.13 | 3.48 |
| | ADA* (Karras et al., 2020a) | 7.40 | 3.55 | 3.05 |
| | **ContraD (ours)** | **7.16** | **3.82** | **2.54** |

## C  RESULTS ON IMAGENET DATASET

Table 8: Comparison of the best FID score and IS on conditional generation of ImageNet (64×64) with BigGAN architecture. Values in the rows marked by * are from those reported in its reference.

| ImageNet | FID $\downarrow$ | IS $\uparrow$ |
|---|---|---|
| BigGAN* (Brock et al., 2019) | 10.6 | $25.43_{\pm 0.15}$ |
| FQ-BigGAN* (Zhao et al., 2020b) | 9.67 | $25.96_{\pm 0.24}$ |
| **ContraD-BigGAN (ours)** | **8.32** | $\textbf{26.94}_{\pm 0.40}$ |

Table 9: Comparison linear evaluation and transfer learning performance across 6 natural image classification datasets for BigGAN discriminators pretrained on ImageNet ($64 \times 64$). We report the top-1 accuracy except for ImageNet and SUN397, which we instead report the top-5 accuracy.

| Training (BigGAN) | ImageNet | CIFAR10 | CIFAR100 | DTD | SUN397 | Flowers | Food |
|---|---|---|---|---|---|---|---|
| Supervised (ImageNet) | 63.5 | 76.1 | 55.2 | 45.4 | 31.7 | 78.1 | 44.5 |
| SimCLR ($\lambda_{\texttt{con}} = \lambda_{\texttt{dis}} = 0$) | 43.4 | 81.2 | 55.3 | 43.9 | 37.6 | 69.8 | 38.8 |
| **ContraD (ours)** | **51.5** | **84.5** | **61.1** | **50.6** | **44.4** | **78.6** | **44.5** |

We also apply ContraD to the state-of-the-art BigGAN (Brock et al., 2019) architecture on ImageNet (Russakovsky et al., 2015), to examine the effectiveness of our method on this large-scale, class-conditional GAN model. For this experiment, we exactly follow the training setups done in FQ-BigGAN (Zhao et al., 2020b), a recent work showing that feature quantization in discriminator could improve state-of-the-art GANs, which is based on the official PyTorch (Paszke et al., 2019) implementation of BigGAN.[4] Following Zhao et al. (2020b), we down-scale the ImageNet dataset into $64 \times 64$, and reduce the default channel width of 96 into 64. We train the model with batch size 512 for 100 epochs ($\sim$250K generator steps). Unlike other experiments, we compute FIDs between the training set and 50K of generated samples for a fair comparison to the baseline scores reported by Zhao et al. (2020b). In Table 8, we show that ContraD further improves the baseline BigGAN training and FQ-BigGAN (Zhao et al., 2020b) both in FID and IS under the same training setups.

Next, we evaluate the representation of the learned BigGAN encoder $D$ from ContraD[5] under linear evaluation (on ImageNet) and several transfer learning benchmarks of natural image classification: namely, we consider CIFAR-10 and CIFAR-100 (Krizhevsky, 2009), the Describable Textures Dataset (DTD) (Cimpoi et al., 2014), SUN397 (Xiao et al., 2010), Oxford 102 Flowers (Flowers) (Nilsback & Zisserman, 2008), and Food-101 (Bossard et al., 2014). As done in Table 3, we compare ContraD with an ablation when $\lambda_{\texttt{con}} = \lambda_{\texttt{dis}} = 0$, which is equivalent to the SimCLR (Chen et al., 2020a) training. In addition, we also compare against the *supervised* baseline, where the representation is pre-trained on ImageNet with standard cross-entropy loss using the same architecture of BigGAN discriminator. Similarly to the ContraD training, this baseline is also trained with the SimCLR augmentation, using Adam optimizer (Kingma & Ba, 2014) for 120 epochs.

Table 9 summarizes the results. We first note that the supervised training achieves 63.5% top-5 accuracy on ImageNet, while the SimCLR baseline achieves 43.4% (in linear evaluation). Again, as also observed in Table 3, ContraD significantly improves the linear evaluation performance from the baseline SimCLR, namely to 51.5%. Although the results are somewhat far behind compared to those reported in the original SimCLR (Chen et al., 2020a), this is possibly due to the use of model with much limited capacity, *i.e.*, the BigGAN discriminator: its essential depth is only $\sim$10, and it also uses spectral normalization (Miyato et al., 2018) for every layer unlike the ResNet-50 architecture considered by Chen et al. (2020a). Nevertheless, we still observe a similar trend in the accuracy gap between supervised and (unsupervised) contrastive learning. On the other hand, the remaining results in Table 9, those from the transfer learning benchmarks, confirm a clear advantage of ContraD, compared to both the SimCLR and supervised baselines: we found the two baselines perform similarly on transfer learning, while our ContraD training further improves SimCLR consistently.

---

[4] https://github.com/ajbrock/BigGAN-PyTorch

[5] We notice here that, although BigGAN is a class-conditional architecture, *i.e.*, it uses the label information in training, the encoder part $D$ in ContraD is still trained in an unsupervised manner, as we put the labels only after stopping gradients, *i.e.*, in the discriminator header $h_{\texttt{dis}}$.

# D QUALITATIVE RESULTS

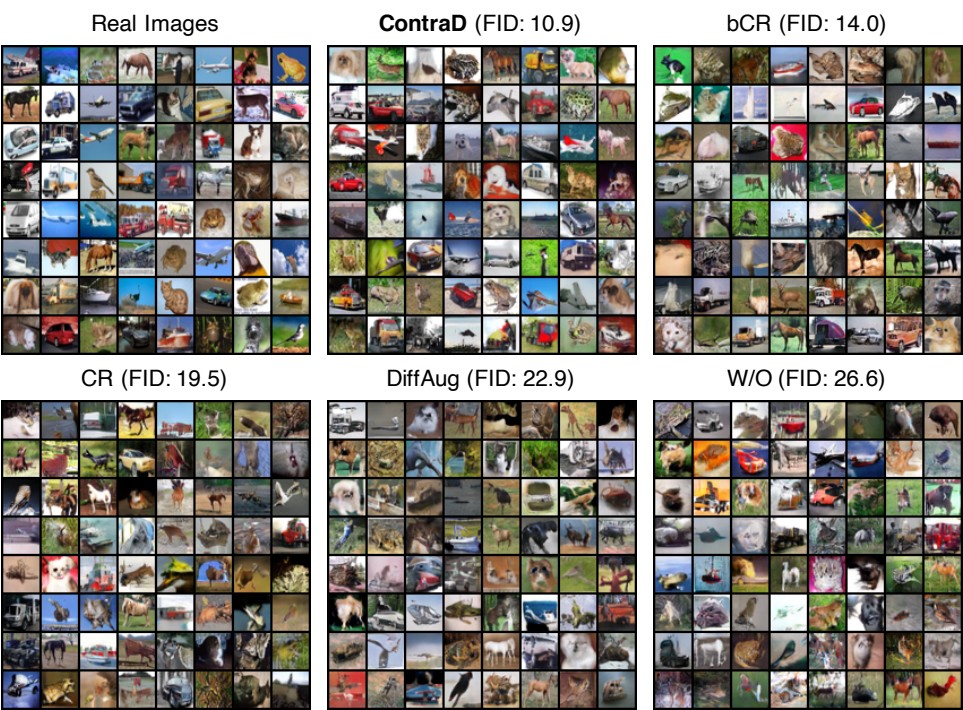

Figure 2: Qualitative comparison of unconditionally generated samples from GANs with different training methods. All the models are trained on CIFAR-10 with an SNDCGAN architecture.

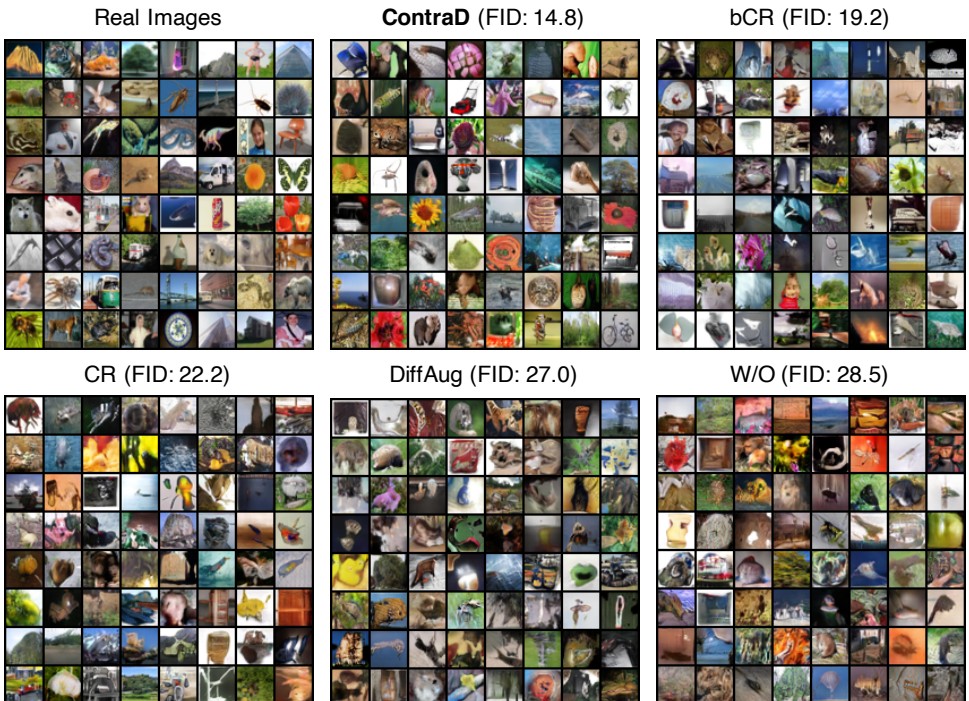

Figure 3: Qualitative comparison of unconditionally generated samples from GANs with different training methods. All the models are trained on CIFAR-100 with an SNDCGAN architecture.

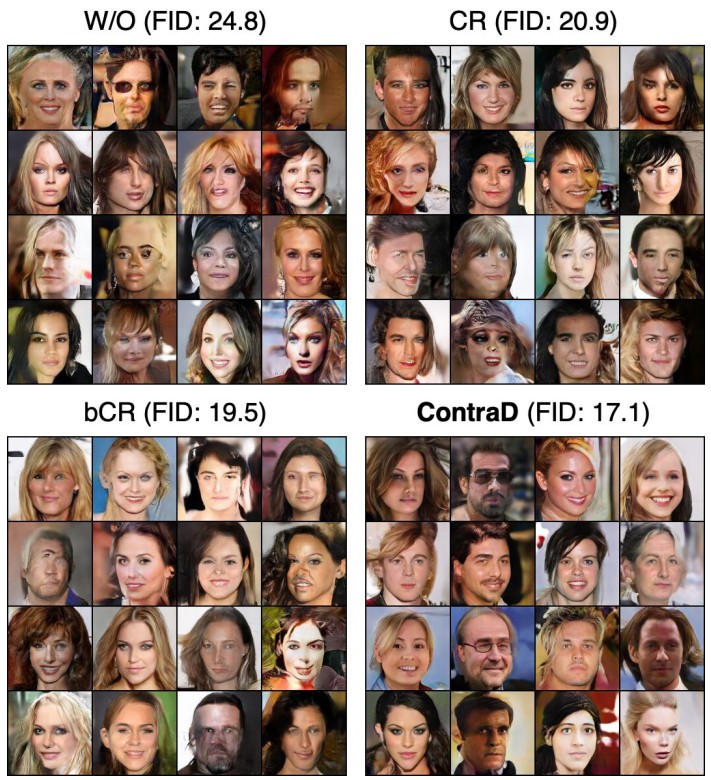

Figure 4: Qualitative comparison of unconditionally generated samples from GANs with different training methods. All the models are trained on CelebA-HQ-128 with an SNDCGAN architecture.

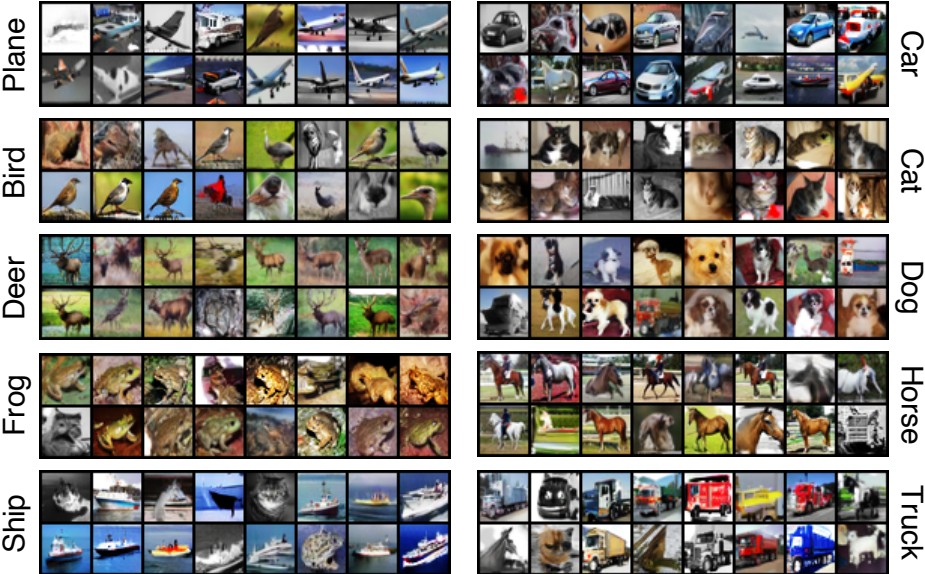

Figure 5: Visualization of conditionally generated samples via conditional DDLS (Section 3.3) with ContraD. We use an SNDCGAN generator trained with an SNResNet-18 ContraD for the generation.

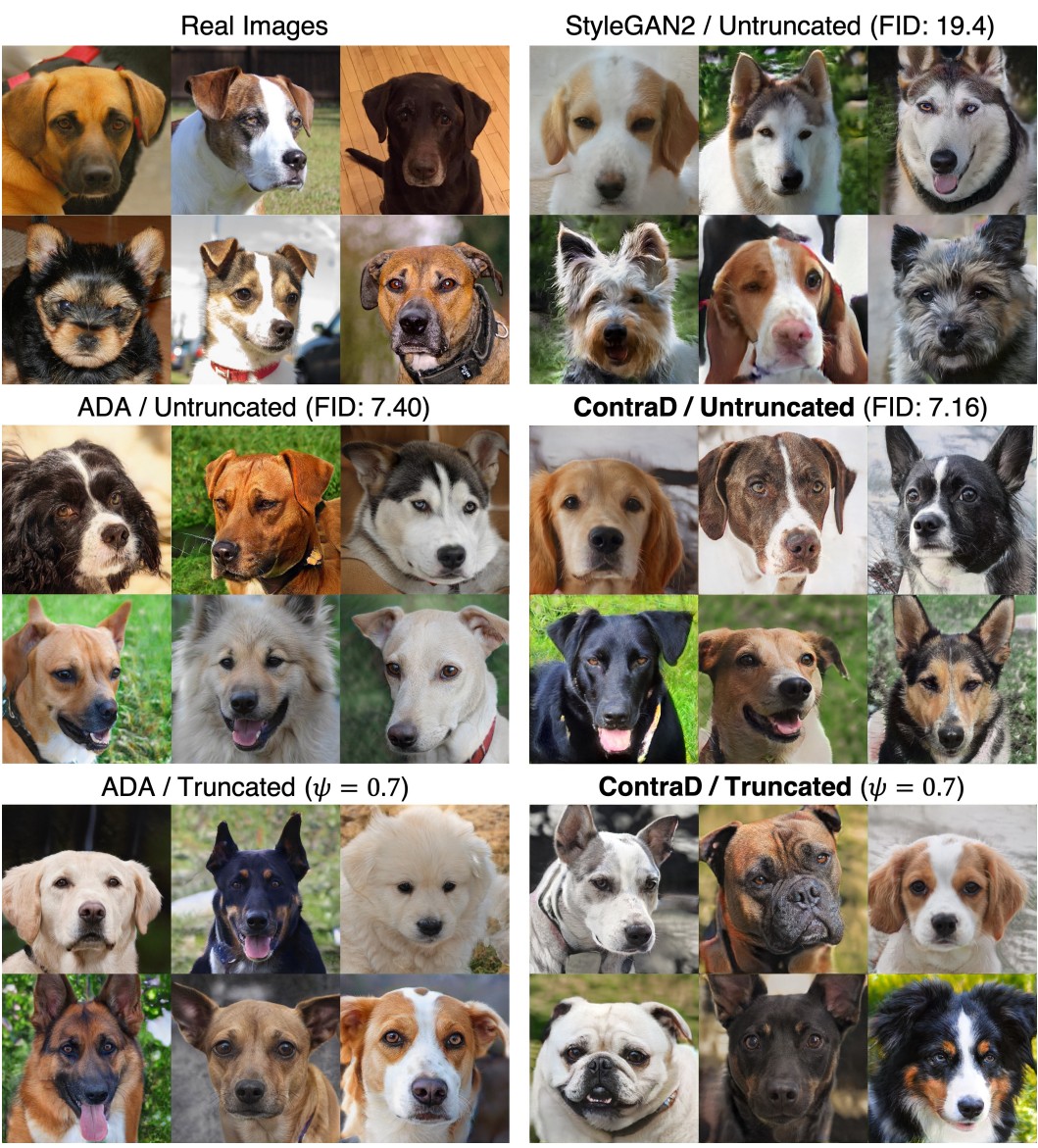

Figure 6: Qualitative comparison of unconditionally generated samples from GANs with different training methods, along with real images from the training set. All the models are trained on AFHQ-Dog (4,739 images) with StyleGAN2. We apply the truncation trick (Karras et al., 2019) with $\psi = 0.7$ to produce the images at the bottom row. We use the pre-trained models officially released by the authors to obtain the results of the baseline StyleGAN2 and ADA.

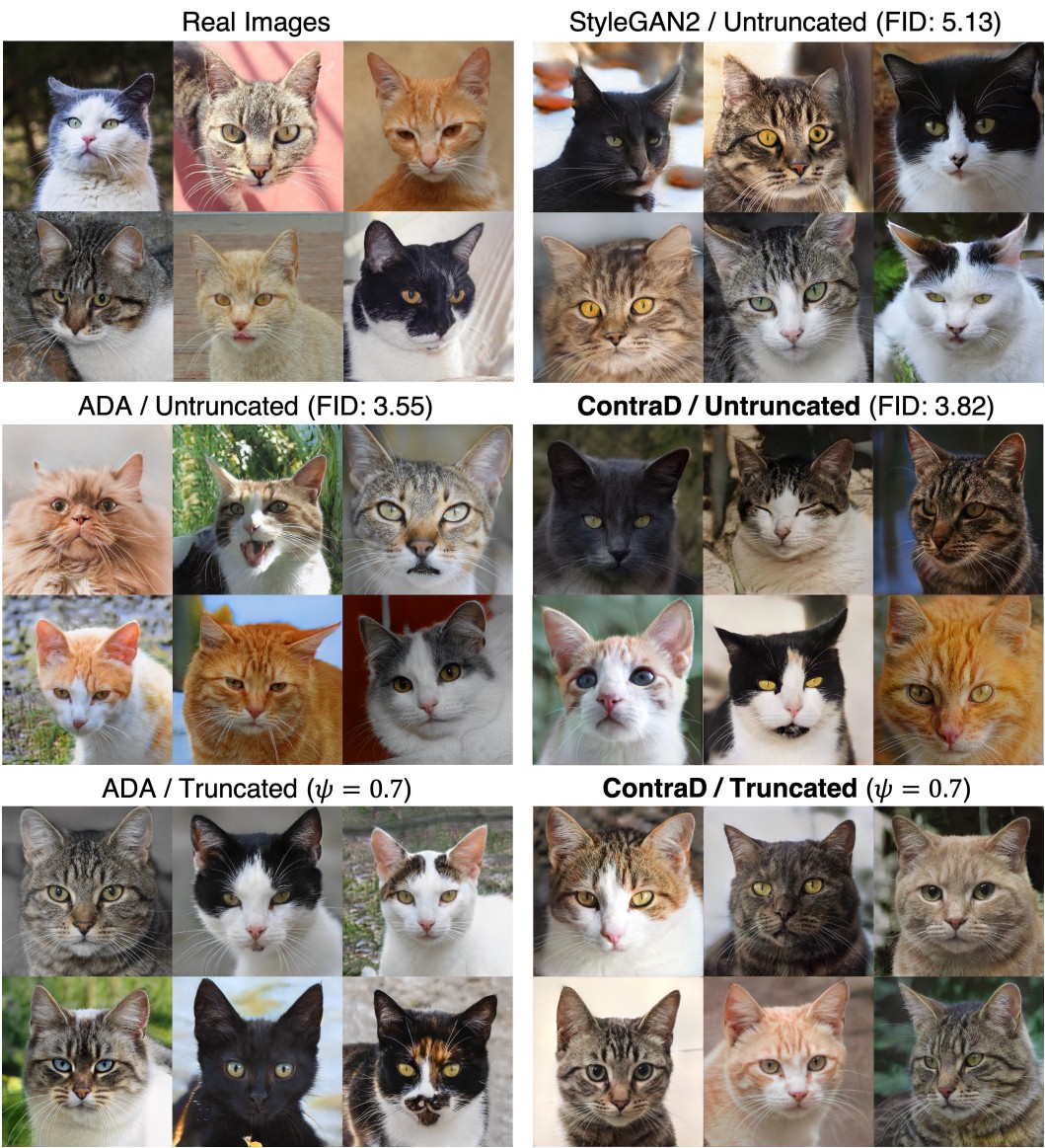

Figure 7: Qualitative comparison of unconditionally generated samples from GANs with different training methods, along with real images from the training set. All the models are trained on AFHQ-Cat (5,153 images) with StyleGAN2. We apply the truncation trick (Karras et al., 2019) with $\psi = 0.7$ to produce the images at the bottom row. We use the pre-trained models officially released by the authors to obtain the results of the baseline StyleGAN2 and ADA.

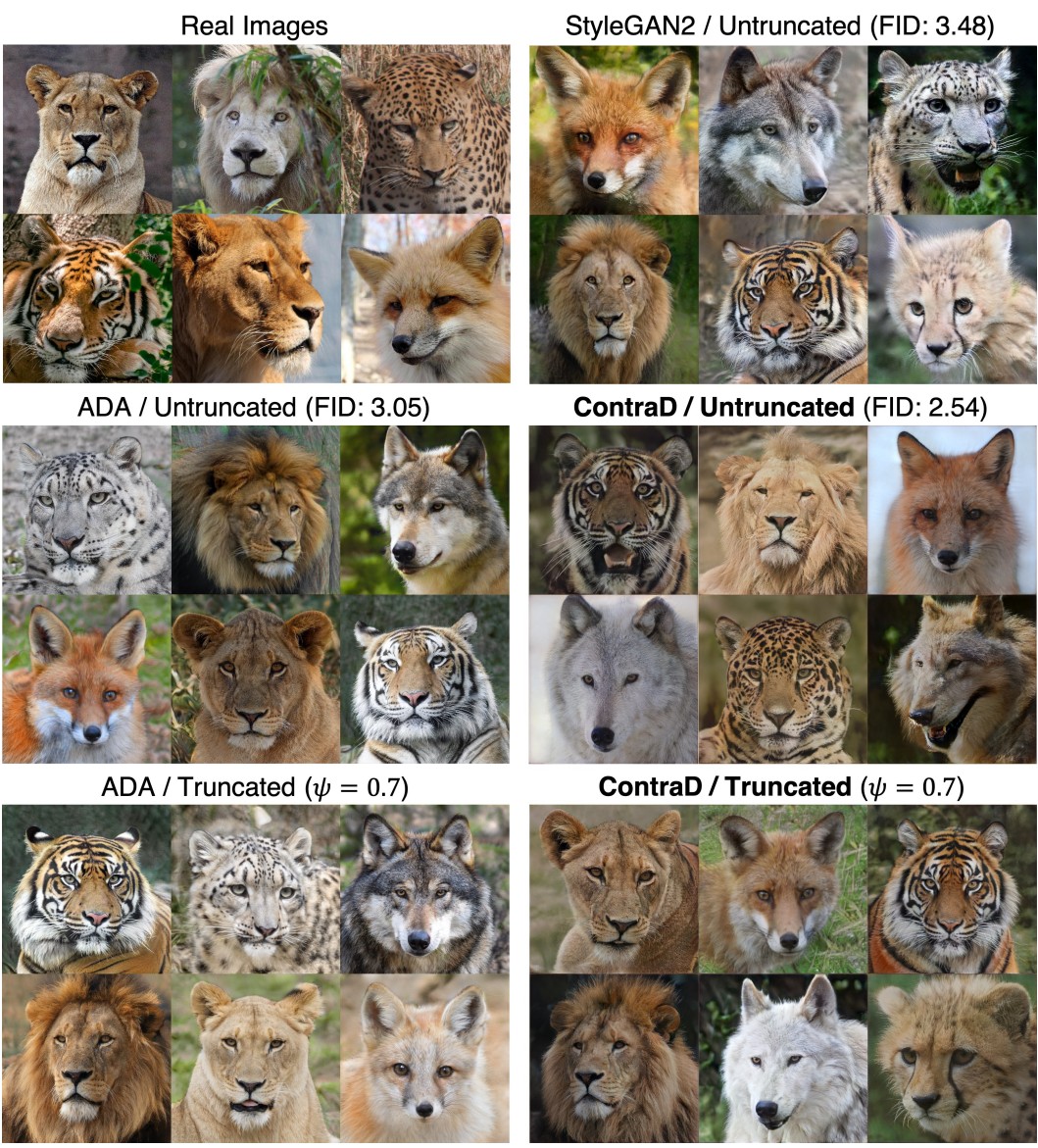

Figure 8: Qualitative comparison of unconditionally generated samples from GANs with different training methods, along with real images from the training set. All the models are trained on AFHQ-Wild (4,738 images) with StyleGAN2. We apply the truncation trick (Karras et al., 2019) with $\psi = 0.7$ to produce the images at the bottom row. We use the pre-trained models officially released by the authors to obtain the results of the baseline StyleGAN2 and ADA.

# E  EFFECT OF USING LARGER BATCH SIZES

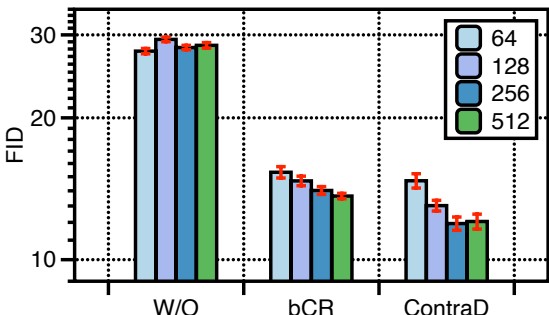

Figure 9: Comparion of the FID distribution of the top 25% of trained models (Kurach et al., 2019) on CIFAR-10 (SNDCGAN) for different batch sizes.

Contrary to existing practices for training GANs, we observe in our experiments that ContraD typically needs a larger batch size in training to perform best. We further confirm this effect of batch sizes in Figure 9: overall, we observe the stanard training ("W/O") is not significantly affected by a particular batch size, while ContraD offers much better FIDs on larger batch sizes. We also observe bCR (Zhao et al., 2020c) slightly benefits from larger batches, but not as much as ContraD does. This observation is consistent with Chen et al. (2020a) that contrastive learning benefits from using larger batch sizes, and it also supports our key hypothesis on the close relationship between contrastive and discriminator representations.

# F  EXPERIMENTAL DETAILS

## F.1  ARCHITECTURE

Overall, we consider four architectures in our experiments: SNDCGAN (Miyato et al., 2018), StyleGAN-2 (Karras et al., 2020b), and BigGAN (Brock et al., 2019) for GAN architectures, and additionally SNResNet-18 for a discriminator architecture only. For all the architectures considered, we have modified the last linear layer into a *multi-layer perceptron* (MLP) having the same input and output size, for a fair comparison to ContraD that requires such an MLP for the discriminator header $h_{\mathrm{d}}$. When a GAN model is trained via ContraD, we additionally introduce two 2-layer MLP projection heads of the output size 128 upon the penutimate representation in the discriminator to stand $h_{\mathrm{r}}$ and $h_{\mathrm{f}}$. All the models are implemented in PyTorch (Paszke et al., 2019) framework.

**SNDCGAN.** We adopt a modified version of SNDCGAN architecture by Kurach et al. (2019) following the experimental setups of other baselines (Zhang et al., 2020; Zhao et al., 2020c). The official implementation can be found in `compare_gan` codebase,[6] and we have re-implemented this TensorFlow implementation into PyTorch framework. The detailed structures of the generator and discriminator of SNDCGAN are summarized in Table 10 and 11, respectively.

**SNResNet-18.** We modify the ResNet-18 (He et al., 2016) architecuture, where a PyTorch implementation is available from `torchvision`[7] library, to not include batch normalization (Ioffe & Szegedy, 2015) layers inside the network. Instead, we apply spectral normalization (Miyato et al., 2018) for all the convolutional and linear layers. We made such a modification to adapt this model as a stable GAN discriminator, as batch normalization layers often harm the GAN dynamics in discriminator in practice.

**StyleGAN2.** We follow the StyleGAN2 architecture used in the DiffAug baseline (Zhao et al., 2020a). Specifically, we generally follow the official StyleGAN2, but the number of channels at 32×32 resolution is reduced to 128 and doubled at each coarser level with a maximum of 512 channels, in order to optimize the model for CIFAR datasets, as suggested by Zhao et al. (2020a).

---

[6]https://www.github.com/google/compare_gan
[7]https://github.com/pytorch/vision/blob/master/torchvision

| Table 10: SNDCGAN generator | | |
| --- | --- | --- |
| Layer | Kernel | Output |
| Latent $\mathbf{z}$ | - | 128 |
| Linear, BN, ReLU | - | $h/8 \times w/8 \times 512$ |
| DeConv, BN, ReLU | $[4, 4, 2]$ | $h/4 \times w/4 \times 256$ |
| DeConv, BN, ReLU | $[4, 4, 2]$ | $h/2 \times w/2 \times 128$ |
| DeConv, BN, ReLU | $[4, 4, 2]$ | $h \quad \times w \quad \times \quad 64$ |
| DeConv, Tanh | $[3, 3, 1]$ | $h \quad \times w \quad \times \quad 3$ |

| Table 11: SNDCGAN discriminator | | |
| --- | --- | --- |
| Layer | Kernel | Output |
| Conv, LeakyReLU | $[3, 3, 1]$ | $h \quad \times w \quad \times \quad 64$ |
| Conv, LeakyReLU | $[4, 4, 2]$ | $h/2 \times w/2 \times 128$ |
| Conv, LeakyReLU | $[3, 3, 1]$ | $h/2 \times w/2 \times 128$ |
| Conv, LeakyReLU | $[4, 4, 2]$ | $h/4 \times w/4 \times 256$ |
| Conv, LeakyReLU | $[3, 3, 1]$ | $h/4 \times w/4 \times 256$ |
| Conv, LeakyReLU | $[4, 4, 2]$ | $h/8 \times w/8 \times 512$ |
| Conv, LeakyReLU | $[3, 3, 1]$ | $h/8 \times w/8 \times 512$ |

**BigGAN.** We use the official PyToch implementation of BigGAN (https://github.com/ajbrock/BigGAN-PyTorch). In order to adapt ContraD into the class-conditional BigGAN architecture, we apply the stop-gradient operation not only on the penultimate representation before computing $h_\mathtt{d}$, but also before applying the class-conditional projection (Miyato & Koyama, 2018).

### F.2 TRAINING DETAILS

**SNDCGAN.** Overall, we follow the best hyperparameter practices explored by Kurach et al. (2019) for SNDCGAN models on CIFAR and CelebA-HQ-128 datasets: We use Adam (Kingma & Ba, 2014) with $(\alpha, \beta_1, \beta_2) = (0.0002, 0.5, 0.999)$ for optimization with batch size of 64, while in case of ContraD on CIFAR datasets the batch size of 512 is used instead. For CelebA-HQ-128, we keep the default batch size of 64 even for ContraD. All models but the "Hinge" baselines in Table 2 are trained with the non-saturating loss (Goodfellow et al., 2014). We stop training after 200K generator updates for CIFAR-10/100 and 100K for CelebA-HQ-128. When CR or bCR is used, we set their regularization strengths $\lambda = 10$ for both real and fake images.

**StyleGAN2.** We follow the training details of DiffAug (Zhao et al., 2020a) in their CIFAR experiments: we use Adam with $(\alpha, \beta_1, \beta_2) = (0.002, 0.0, 0.99)$ for optimization with batch size of 32, but 64 for ContraD models. We use non-saturating loss for training, and use $R_1$ regularization (Mescheder et al., 2018) with $\gamma = 0.1$. We do not use, however, the path length regularization and the lazy regularization (Karras et al., 2020b) in training. We take exponential moving average on the generator weights with half-life of $10^6$. We stop training after 800K generator updates. When CR or bCR is used, we set their regularization strengths $\lambda = 10$ for both real and fake images.

**Linear evaluation and transfer learning.** For a single linear evaluation (and transfer learning) run, we train a linear classifier upon a given (frozen) representation. The results in Table 3 are trained via stochastic gradient descent for 100 epochs: the initial learning rate of 0.1, and it is decayed by 0.1 at 60, 75, and 90-th epoch. In case of Table 9, on the other hand, we use Adam optimizer (Kingma & Ba, 2014) for 90 epochs: the initial learning rate if 0.0002, and it is decayed by 0.3 at 60 and 75-th epoch. We augment every training dataset via random crop, resizing and horizontal flipping.

**Conditional DDLS.** We run Langevin sampling of step size $\varepsilon = 0.1$ for 1,000 iterations. We follow all the practical considerations proposed by Che et al. (2020) to improve the sample quality of DDLS: specifically, (a) we separately set the standard deviation of the Gaussian noise (13) to 0.1, and (b) to alleviate mode dropping issues, instead of running DDLS on $G$ itself, we run the sampling with $G^*(\mathbf{z}, \mathbf{z}') := G(\mathbf{z}) + \varepsilon \mathbf{z}'$, where $\mathbf{z}'$ is also jointly updated via Langevin dynamics.

**Hyperparameters in ContraD.** Unless otherwise specified, we use $\lambda_\mathtt{con} = \lambda_\mathtt{dis} = 1$, and $\tau = 0.1$ in our experiments, where $\tau$ is the temperature hyperparameter defined in (5). We use hidden layer of size 512 for $h_\mathtt{r}, h_\mathtt{f}, h_\mathtt{d}$ in SNDCGAN and StyleGAN2, and 1024 in SNResNet-18 and BigGAN. For ContraD models, we use linear warm-up strategy on learning rate up to 3K steps of training for both generator and discriminator, following practices in contrastive learning (Chen et al., 2020a).

**SimCLR augmentations.** We use the augmentation pipeline proposed in SimCLR (Chen et al., 2020a) for training ContraD. Specifically, for CIFAR datasets, we sequentially transform a given image by: (a) random crop and resizing, (b) horizontal flipping, (c) color jittering with probability of $p = 0.8$, and (d) graying the image with probability of $p = 0.2$. We additionally apply (f) Gaussian blurring with probability of $p = 0.5$ for higher resolution images such as CelebA-HQ-128 and ImageNet. We ensure that every transformation in this pipelines are differentiably implemented, so that the augmentation still can be used for training the generator of GAN.

