# OpenReview forum: "Training GANs with Stronger Augmentations via Contrastive Discriminator"
_ICLR.cc/2021/Conference — ICLR 2021 Poster_

### Official Review · AnonReviewer2 · 2020-10-28
**A combination of GAN and contrastive learning, but with unaddressed concerns**

**Rating:** 6
**Confidence:** 4

**Review:**

The authors propose to improve GAN training by incorporating augmentations from contrastive learning. Specifically, a new contrastive discriminator, named ContraD, is proposed for GANs; with ContraD, the encoder part of the discriminator is trained with (two) contrastive learning losses, while the left discriminator head and the GAN generator are trained as usual. The authors argue that this specific fusion of GAN and contrastive learning signiﬁcantly stabilizes GAN training and, moreover, the fused two research fields could benefit each other.

The paper is well written overall. The idea is interesting, and the technical details may be sound. However, I am not fully convinced because of the concerns listed below.

Although the approach is proposed to stabilize GAN training, this aspect was not highlighted in the experiments.

In Section 3.1, the discriminator encoder is trained with two specifically-chosen self-supervised-learning losses, and the experiments show that such a specific combination is essential for a good performance; otherwise, “the GAN loss could completely negate the effectiveness of ContraD. ” So why would that happen? How to choose those losses in practice? Detailed discussions are necessary here.

The first sentence of Section 3.2 is contradictory with Eq (12) and Algorithm 1.

In Eq (14), how to choose the v for a specific class? Was the label information used to get the v?

In Table 1, it seems the performance of StyleGAN2 is much lower than that reported in the original paper. Please elaborate on that.

In the experiments, larger batch sizes are used for the proposed method. Will that affect the fairness? Discussions are necessary.

---

> ### Author Response · Authors · 2020-11-16
> **Response to R2**
>
> We sincerely appreciate your thoughtful comments, efforts, and time. We respond to each of your questions and concerns one-by-one in what follows: possibly, some questions that need additional expensive experiments to address may not be included in the current response, but they will be supplemented after a revised manuscript is updated. We hope to deliver the revision soon, where we will make sure to reflect all comments.
>
> ---
> **Q1. "Although the approach is proposed to stabilize GAN training, this aspect was not highlighted in the experiments."**
>
> We use the term "stability" in a broad sense, e.g., our experiments view this in terms of the final performance of GAN after the training: the high-fidelity (e.g. high IS), yet diverse (e.g. low FID) generation from a GAN cannot be achieved with an instability (e.g., mode collapse) in training. Nevertheless, we agree that the term of stability in GAN training might be confusing for some readers, and we will clarify this in the revision.
>
> ---
> **Q2. "... the GAN loss could completely negate the effectiveness of ContraD. ...": why does it happen?**
>
> Recall that the encoder $D$ of ContraD minimizes two losses: namely, $L_{\tt con}^{+}$ and $L_{\tt con}^{-}$ (but not $L_{\tt dis}$), and $L_{\tt con}^{+}$ is simply the SimCLR loss [Chen et al., 2020] (as mentioned in Section 3.1). Here, the referred line is to emphasize the importance of choosing $L_{\tt con}^{-}$: the role of $L_{\tt con}^{-}$ is (a) forcing $D$ to be able to discriminate real vs. fake samples, while (b) not compromising the representation from $L_{\tt con}^{+}$. One of our findings is that our proposed supervised contrastive form (Eq. 8) of loss does satisfy both of (a) and (b): the pure GAN loss as $L_{\tt con}^{-}$ may satisfy (a), but not (b).
>
> ---
> **Q3. "The first sentence of Section 3.2 is contradictory with Eq. 12 and Algorithm 1."**
>
> Thank you for pointing out this detail. We note that the generator loss of ContraD (Eq. 12) is indeed equivalent to the original non-saturating loss if we regard $\sigma(h_{\tt d}(D(\cdot)))$ as a single discriminator. The difference in formula occurs since we re-define $D$ to be vector-valued (Section 3.1) to explain ContraD. We will clarify this in the revision.
>
> ---
> **Q4. How to choose the $v$ in Eq. 14 for a specific class? Was the label information used to get the $v$?**
>
> In our experiments, we do use the label information to find a class-wise $v$: namely, we simply take the linear weights learned from a fine-tuning upon the ContraD representation  (e.g. linear evaluation protocol), as also mentioned in p.7. Nevertheless, in general, the notion of $v$ is independent to labels, e.g., one can apply an unsupervised clustering method on ContraD to obtain another $v$. An important point here is that such a label information is completely unused when training the base ContraD.
>
> ---
> **Q5. "The performance of StyleGAN2 in Table 1 is much lower than that reported in the original paper [Karras et al., 2020]?"**
>
> To the best of our knowledge, the original StyleGAN2 paper [Karras et al., 2020] does not perform CIFAR-10/100 experiments. Hence, the reported CIFAR-10/100 experimental results (and detailed setups) for StyleGAN2 are instead from DiffAug [Zhao et al., 2020] (Appendix E for the details) for a fair comparison in Table 1, where we have confirmed the reproducibility of them.
>
> ---
> **Q6. Larger batch sizes for ContraD - will that affect the fairness?**
>
> We actually investigated the detailed effect of batch sizes on the standard, bCR, and ContraD training in Appendix D, confirming that the benefit from larger batch sizes is not common in GAN training: e.g., we observe the standard GAN training works best at the smallest batch size of 64, which is also consistent with the popular practices in GAN [Lucic et al., 2018; Kurach et al., 2019]. As mentioned in Section 4, the choice of larger batch for ContraD rather comes from integrating SimCLR into GAN, which is our unique perspective.
>
> ---
> - [Lucic et al., 2018] Are GANs Created Equal? A Large-Scale Study, NeurIPS 2018.
> - [Kurach et al., 2019] A Large-Scale Study on Regularization and Normalization in GANs, ICML 2019.
> - [Chen et al., 2020] A Simple Framework for Contrastive Learning of Visual Representations, ICML 2020.
> - [Karras et al., 2020] Analyzing and Improving the Image Quality of StyleGAN, CVPR 2020.
> - [Zhao et al., 2020] Differentiable Augmentation for Data-Efficient GAN Training, NeurIPS 2020.

---

> > ### Comment · AnonReviewer2 · 2020-11-23
> > **On Q5, the performance over the work based on StyleGAN2  [karras2020training]**
> >
> > I have made a mistake about the reference paper, sorry. Actually, I mean the work based on StyleGAN2  [karras2020training] has shown strong performance on CIFAR10 with FID=2.67 and IS=10.06, which is much better than the performance show in table 1, please elaborate on that.
> >
> > [karras2020training] Training Generative Adversarial Networks with Limited Data

---

> > > ### Author Response · Authors · 2020-11-23
> > > **Response to R2 (2)**
> > >
> > > Many thanks for clarifying your question before the discussion phase ends.
> > >
> > > As mentioned in the previous response, we follow the StyleGAN2 experiment setup of DiffAug [Zhao et al., 2020] that is a highly concurrent work of ADA [Karras et al., 2020] that you are referring to.
> > >
> > > Our results in Table 1 (and DiffAug) can be lower than those reported in ADA [Karras et al., 2020], as the detailed experimental setups are largely different in many aspects:
> > >
> > > - (a) - ADA uses 4x longer training than ours (and DiffAug) with many different hyperparameters, e.g., larger learning rate. The actual architectures are slightly different as well: ADA uses a larger one.
> > > - (b) - ADA computes FIDs differently to ours (and DiffAug). More specifically, we compute FID of 10K generated samples vs. 10K "test" samples of CIFAR, while ADA does of 50k generated samples vs. all available real samples, i.e., 60K "train+test" samples: FID tends to be smaller on larger sample sizes, as done in ADA. Nevertheless, we believe our criteria of using only test samples is a more standard and reasonable way of computing FIDs, as also argued in prior works [Lucic et al., 2018; Kurach et al., 2019].
> > > - (c) - The two concurrent methods, ADA and DiffAug, are very similar, except for ADA’s additional "dynamic" augmentation scheme driven from validation samples: i.e., the setup would assume an extra split for the validation. As also mentioned (as a footnote) in p.6, this is another reason why we compare DiffAug instead of ADA for a clearer evaluation: our goal is to verify which method better incorporates a given (fixed) augmentation.
> > >
> > > Hence, especially due to (b) and (c), we have decided to follow DiffAug rather than ADA for designing our experimental setups, and think our current evaluation does verify the effectiveness of our method on StyleGAN2. However, if you think experiments under the ADA’s setup are valuable to add, we will incorporate the results in the final draft (please understand that it will not be possible to update them until the end of the discussion period, i.e., in 2 days, considering that these experiments would roughly take ~1000 GPU*hrs for each training configuration).
> > >
> > > ---
> > > - [Lucic et al., 2018] Are GANs Created Equal? A Large-Scale Study, NeurIPS 2018.
> > > - [Kurach et al., 2019] A Large-Scale Study on Regularization and Normalization in GANs, ICML 2019.
> > > - [Zhao et al., 2020] Differentiable Augmentation for Data-Efficient GAN Training, NeurIPS 2020.
> > > - [Karras et al., 2020] Training Generative Adversarial Networks with Limited Data, NeurIPS 2020.

---

> > > > ### Comment · AnonReviewer2 · 2020-11-24
> > > > **Thanks for the response**
> > > >
> > > > Thanks for the detailed response. It will be great value if you can take the experiments under the ADA’s setup into consideration, since ADA is also claimed to apply a wide range of augmentations. At least, a discussion (advantages and disadvantages) on comparing these two methods will be helpful for audience.

---

> > > > > ### Author Response · Authors · 2020-11-24
> > > > > **Response to R2 (3)**
> > > > >
> > > > > Thanks for your thoughtful suggestion. We will include a more detailed discussion or experiments on ADA compared to our method in the final draft. We also express our gratitude for reconsidering your score on our manuscript!

---

### Official Review · AnonReviewer4 · 2020-10-29

**Rating:** 6
**Confidence:** 4

**Review:**

In this paper, the authors suggest using the contrastive loss to improve the training of the discriminator and further stabilize the GAN training process. More specifically, the proposed method incorporates the self-supervised simCLR contrastive loss on a pair of transformed real images and supervised contrastive loss on the fake ones. The proposed method is evaluated on the image synthesis task on CIFAR10/100 and CelebA-HQ-128 images and over several different GAN models.

Strengths:
* The idea of using self-supervised learning for improving the training dynamics of the discriminator makes sense and is an interesting exploration area.
* Empirical evaluations show a consistent and significant advantage for the proposed methods and ablation studies verify the contributions of the different proposed components.

weaknesses:
* The proposed method is rather a careful ensembling of existing components, e.g. simCLR self-supervised or supervised contrastive loss in the right context, rather than a radically novel methodological contribution.
* The proposed method is only tested on relatively low-resolution datasets, namely CIFAR10/100 and CelebA-HQ-128. It would have been interesting to also demonstrate the contributions on the more challenging higher resolution datasets.

Detailed comments:
* Equation 7: If I am not mistaken, it doesn't seem quite the same as in the referred supervised contrastive loss from Khosla et al.; more specifically, I think the order between log and \sum_{v_{i+}^{(2)} \in V_{i+}^{(2)}} need to be reversed. Please clarify.
* An ablation study I was missing was having the proposed method without the extensive simCLR augmentations and see how it compares to the other methods.
* It would have been interesting to also compare with some other recent and relevant work, e.g. ADA (Karras et al.).
* "Remark that we use an independent projection header h_f instead of h_r" => Is this making a significant difference? Are there ablation studies showing this?
* Typo: "approaches that handles"

---

> ### Author Response · Authors · 2020-11-16
> **Response to R4 (1/2)**
>
> We sincerely appreciate your thoughtful comments, efforts, and time. We respond to each of your questions and concerns one-by-one in what follows: possibly, some questions that need additional expensive experiments to address may not be included in the current response, but they will be supplemented after a revised manuscript is updated. We hope to deliver the revision soon, where we will make sure to reflect all comments.
>
> ---
> **Q1. Methodological contributions**
>
> We first note that GAN and SimCLR have been two independently-developed methods, and combining them as a new working method can be highly non-trivial: even, there is a challenge in engineering them, as the popular experimental setups (e.g., the choice of neural architectures or batch sizes) for GAN and SimCLR are different and their performance is highly sensitive to them.
>
> Our key ideas have enabled the successful combination of GAN and SimCLR (as supported in our ablation study of Section 4.2), as well as they give many insights to both techniques: for example, our new design of introducing a small header to minimize the GAN loss upon other (e.g., contrastive) representation is a promising yet unexplored way of designing a new GAN architecture. For the SimCLR side, on the other hand, we suggest a new idea of incorporating "fake" samples for contrastive learning, also an interesting direction along with other recent attempts to improve the efficiency of negative sampling in contrastive learning, e.g., via hard negative mining [Kalantidis et al., 2020; Robinson et al., 2020]. Overall, we do believe that our work sheds a new angle to both worlds, GAN and SimCLR.
>
> We will add respective comments to the revision.
>
> ---
> **Q2. Results on more challenging datasets?**
>
> For your information, we note here that our experiments have also covered ImageNet with BigGAN architecture [Brock et al., 2019] in Appendix B. Nevertheless, following your suggestion, we will continue to work on additional higher resolution datasets and deliver their results in the revision (if time permits) or the final draft.
>
> ---
> **Q3. ContraD without SimCLR augmentation?**
>
> Following the suggestion made by you (and R1), we will update Table 5 in the revision to include an additional ablation of ContraD with a weaker augmentation of "HFlip, Trans": as summarized below, we observe that ContraD is still as good as (or better in terms of IS) a strong baseline of "balanced consistency regularization" (bCR) [Zhao et al., 2020]. The performance degradation compared to "ContraD + SimCLR" is due to that SimCLR requires strong augmentations to learn a good representation [Chen et al., 2020], i.e., ContraD with weaker augmentations learns worse representation in terms of its linear evaluation performance as shown in the following table.
>
>
> | Method  |   Augment.   | FID $\downarrow$ |  IS $\uparrow$ | Lin. Eval. (%) |
> |:--------|:------------:|:----:|:----:|:--------------:|
> | -       | -            | 26.6 | 7.38 |       -        |
> | bCR	    | HFlip, Trans | 14.0 | 8.35 |       -        |
> | bCR + Aug.  | SimCLR       | 20.6 | 7.44 |       -        |
> | ContraD | SimCLR       | 10.9 | 8.78 |  77.5 $\pm$ 0.20  |
> | ContraD - Aug.  | HFlip, Trans | 13.7 | 8.54 |  72.9 $\pm$ 0.04  |
>
> ---
> **Q4. Comparison with ADA [Karras et al., 2020]?**
>
> As mentioned (as a footnote) in p.6, we do not compare with ADA [Karras et al., 2020] but do compare with its concurrent work, namely DiffAug [Zhao et al., 2020]. It is mainly for a clearer comparison of which method better incorporates a given (fixed) augmentation: ADA offers a similar method to DiffAug, but with an additional "dynamic" augmentation scheme using a validation dataset, which can be viewed as an orthogonal idea to our method.
>
> ---
> **Q5. What if $h_f = h_r$?**
>
> We have empirically observed that having separate projection headers for $h_f$ and $h_r$ is empirically more stable to train than sharing them: e.g., under $h_f = h_r$, we could not obtain a reasonable performance with ContraD at "G: SNDCGAN / D: SNResNet-18" on CIFAR-10 (FID ($\downarrow$) / IS ($\uparrow$) = 194.6 / 3.02). Intuitively, an optimal embedding (after projection) from the supervised contrastive loss ($L_{\tt con}^{-}$) can be harmful to those from SimCLR, as the former would encourage the embeddings of real samples to shrink into a single point, which is, in some sense, the opposite direction of SimCLR. We will add the respective discussion and results in the revision.

---

> > ### Author Response · Authors · 2020-11-16
> > **Response to R4 (2/2)**
> >
> > ---
> > **Q6. Editorial comments**
> >
> > Many thanks for the careful reading, especially for pointing out an important typo in Eq. 7: as suggested, the order of log and sum must be reversed. Your editorial comments are now all fixed in the revised manuscript.
> >
> > ---
> > - [Brock et al., 2019] Large Scale GAN Training for High Fidelity Natural Image Synthesis, ICLR 2019.
> > - [Chen et al., 2020] A Simple Framework for Contrastive Learning of Visual Representations, ICML 2020.
> > - [Kalantidis et al., 2020] Hard Negative Mixing for Contrastive Learning, NeurIPS 2020.
> > - [Karras et al., 2020] Training Generative Adversarial Networks with Limited Data, NeurIPS 2020.
> > - [Robinson et al., 2020] Contrastive Learning with Hard Negative Samples, 2020.
> > - [Zhao et al., 2020] Differentiable Augmentation for Data-Efficient GAN Training, NeurIPS 2020.

---

> > > ### Comment · AnonReviewer4 · 2020-11-23
> > >
> > > I appreciate the detailed responses that the authors have provided to the questions and concerns I raised. I am particularly satisfied with the new experiments conducted to address questions 3 and 5, providing further evidence for some of the claims made by the authors.
> > > I hope further experiments on higher resolution datasets will obtain similar conclusions.
> > >
> > > Regarding the raised concern about the methodological novelties, I agree with the reviewers that the proposed combination is novel, but I still think this is not sufficient to consider strong methodological novelties as a major strength for this paper.

---

> > > > ### Author Response · Authors · 2020-11-24
> > > > **Response to R4 (2)**
> > > >
> > > > Thank you for your positive response before the discussion phase ends, and we are happy to hear that our response could help to address your questions!
> > > >
> > > > Indeed, we are currently working on higher resolution datasets as you suggested, and believe these additional results would greatly strengthen our paper. We hope for your understanding that we could not update the results in this phase due to limited time and resources, and we are willing to add them in the final draft.

---

### Official Review · AnonReviewer3 · 2020-10-29
**Good work on synthesizing self-supervised training and GANs; but some claims are perhaps too strong**

**Rating:** 7
**Confidence:** 3

**Review:**

**Summary**
The manuscript proposes ContraD - a method that incorporates the recent SimCLR self-supervised learning method for images into the GAN training framework. Experimental results show that the proposed method consistently improves on strong baselines in terms of FID scores.

**Score justification**
The paper is well-written, the introduction carefully synthesizes a lot of recent work on self-supervised representation learning and GANs; results are convincing (albeit mostly obtained on smaller-scale datasets) and ablations support paper claims.


**Major comments**
* Ablation of $L_{con}^{+}$ in Table 6 tells a mixed story - it breaks down for the SNResNet discriminator, but seems to work well for the SNDCGAN discriminator. It's hard to conclude something definitive from the failed SNResNet experiment (could have worked with more hyper-parameter tuning?), but the successful SNResNet experiment suggests that the $L_{con}^{+}$ may not be necessary at all, which is at odds with the authors' narrative. Would be great to see more ablations showing that this is not the case, including a combined ablation of $L_{con}^{+}$ , $L_{con}^{-}$ and stop gradients.

* It would be interesting to see linear separation and transfer to other datasets (as in the SimCLR paper) for the ImageNet models. Strong results there would support the authors' narrative of the synergy GANs and SimCLR training.
* I am not fully convinced by the strong claims of strong coherence between GANs and contrastive representation learning (e.g. "[...] these two representations greatly complement each other [...]", "[...] a good contrastive representation is good for GANs and vise versa"). In particular it seems to be that GAN training benefits from contrastive learning (it seems to be a good auxiliary loss in presence of strong augmentation), but the claims of benefit to SimCLR don't seem fully substantiated.

**Minor comments**
* Multiple citations do not have correct capitalization (e.g. "gan")
* Results presented in Table 3 (linear evaluation on CIFAR-10) can be a bit misleading - at first glance they seem to suggest that * ContraD improves on SimCLR results from the original paper (model pre-trained on ImageNet and transferred to CIFAR-10). Making it clear that this is not the case (e.g. by including the original SimCLR results and/or supervised performance) would be helpful to the readers.
* Similarly, Table 4 would be easier to take in in the context of FID scores that are achieved by class-conditional GANs.

---

> ### Author Response · Authors · 2020-11-16
> **Response to R3**
>
> We sincerely appreciate your thoughtful comments, efforts, and time. We respond to each of your questions and concerns one-by-one in what follows: possibly, some questions that need additional expensive experiments to address may not be included in the current response, but they will be supplemented after a revised manuscript is updated. We hope to deliver the revision soon, where we will make sure to reflect all comments.
>
> ---
> **Q1. $L_{\tt con}^{+}$ may not be necessary, if there exists a better hyperparameter that works at ("No $L_{\tt con}^{+}$", SNResNet-18) in Table 6?**
>
> Many thanks for an insightful comment. Our focus is to improve both GAN and SimCLR in a unified angle, where all our experiments are designed to support this. $L_{\tt con}^{+}$ is the essential component of SimCLR and without this, it would not be possible to perform linear evaluation (Table 3) or conditional generation (Table 4) upon the ContraD representation in our experiments.
>
> Even though SNDCGAN could be trained without $L_{\tt con}^{+}$ as you pointed out, using $L_{\tt con}^{+}$ clearly improves its performance further as shown in Table 6: i.e., even if one finds other good hyperparameters for SNResNet, $L_{\tt con}^{+}$ is still useful. We also remark the hyperparameter used in our experiments is a popular one searched from a large-scale study [Lucic et al., 2018], which is known to widely work in practice: SNResNet without $L_{\tt con}^{+}$ has failed in this setup, but somewhat interestingly, $L_{\tt con}^{+}$ remedies the issue.
>
> ---
> **Q2. Linear evaluation and transfer learning on the ImageNet models**
>
> Following your suggestion, we will report the results in the revision.
>
> ---
> **Q3. Not fully convinced to the claim "ContraD is beneficial to SimCLR"**
>
> We agree that we put more efforts in evaluation and justification on why SimCLR is beneficial to GAN. As a byproduct, we also observe the opposite direction is possible from the empirical results in Table 3: although they are conducted in smaller scale than the original setup of SimCLR [Chen et al., 2020] (which is based on ImageNet with ResNet-50), our results clearly confirm that $L_{\tt con}^{-}$ with another network $G$ (which is trained via $L_{\tt dis}$ and $L_G$) is an effective auxiliary loss for SimCLR training. This effectiveness of "fake" samples in contrastive learning has been unexplored, but we think it is promising along with the recent line of research to improve the efficiency of negative sampling in contrastive learning, e.g., via hard negative mining [Kalantidis et al., 2020; Robinson et al., 2020]. In this respect, we believe scaling up ContraD to compete with other state-of-the-art self-supervised learning benchmarks is an important future direction. We will provide such a more detailed discussion in the revision.
>
> ---
> **Q4. Other minor comments**
>
> Thank you for making constructive suggestions to improve the clarity of our manuscript. We will incorporate them in the revision.
>
> ---
> - [Lucic et al., 2018] Are GANs Created Equal? A Large-Scale Study, NeurIPS 2018.
> - [Chen et al., 2020] A Simple Framework for Contrastive Learning of Visual Representations, ICML 2020.
> - [Kalantidis et al., 2020] Hard Negative Mixing for Contrastive Learning, NeurIPS 2020.
> - [Robinson et al., 2020] Contrastive Learning with Hard Negative Samples, 2020.

---

### Official Review · AnonReviewer1 · 2020-11-03
**a well-executed empirical paper with strong performance**

**Rating:** 7
**Confidence:** 4

**Review:**

==== Summary ====

This paper improves upon state-of-the-art GANs by incorporating recent advances of contrastive representation learning into the training of discriminator. In particular, the discriminator loss function consists of three terms: (1) the original SimCLR loss on the multi-view real data pairs; (2) the supervised contrastive loss (Khosla et al, 2020) that assigns high scores among the fake sample pairs and giving lower score among the real data pairs; (3) the usual discriminator loss in GAN training. While each of these terms alone is not entirely new, the author proposes several tricks to make the training of GANs together with the contrastive loss works. Empirically, the proposed method outperforms other GAN methods trained with auxiliary data augmentation techniques, and demonstrates good representations under the linear classifier probing setup.

Pros:

(1) Writing is clear and easy to follow

(2) Strong empirical performance in both image generation and classifier probing
Solid experiment designs with thorough ablation studies

Cons:

(1) Rather limited novelty in terms of technical contributions

==== Technical Questions ====

Q1: Regarding the supervised contrastive loss for fake samples (i.e., L_{con}^{-} in Figure 1).
I am wondering if it is beneficial to also consider data augmentation on the fake sample produced by generators. In that case, the loss function matrix in Figure 1 will look closer to the L_{con}^{+} part, which encourages different views of the same fake sample to have similar representations.

Q2: I feel like there’s one ablation setting missing in Table 1: using “HFlip, Trans” data augmentation for the ContraD. This way, we can see the true benefit of combining those three losses.

Q3: For the linear evaluation results in Table 3, SimCLR results seem not consistent with their original paper? What causes the difference?

---

> ### Author Response · Authors · 2020-11-16
> **Response to R1**
>
> We sincerely appreciate your thoughtful comments, efforts, and time. We respond to each of your questions and concerns one-by-one in what follows: possibly, some questions that need additional expensive experiments to address may not be included in the current response, but they will be supplemented after a revised manuscript is updated. We hope to deliver the revision soon, where we will make sure to reflect all comments.
>
> ---
> **Q1. Limited technical contributions**
>
> We first note that GAN and SimCLR have been two independently-developed methods, and combining them as a new working method can be highly non-trivial: even, there is a challenge in engineering them, as the popular experimental setups (e.g., the choice of neural architectures or batch sizes) for GAN and SimCLR are different and their performance is highly sensitive to them.
>
> Our key ideas have enabled the successful combination of GAN and SimCLR (as supported in our ablation study of Section 4.2), as well as they give many insights to both techniques: for example, our new design of introducing a small header to minimize the GAN loss upon other (e.g., contrastive) representation is a promising yet unexplored way of designing a new GAN architecture. For the SimCLR side, on the other hand, we suggest a new idea of incorporating "fake" samples for contrastive learning, also an interesting direction along with other recent attempts to improve the efficiency of negative sampling in contrastive learning, e.g., via hard negative mining [Kalantidis et al., 2020; Robinson et al., 2020]. Overall, we do believe that our work sheds a new angle to both worlds, GAN and SimCLR.
>
> We will add respective comments to the revision.
>
> ---
> **Q2. What if $L_{\tt con}^{-}$ uses another view of fake samples, in a similar manner to $L_{\tt con}^{+}$?**
>
> Many thanks for an insightful comment. We have actually tried several possible variants of $L_{\tt con}^{-}$ including “multi-view” losses as you suggested, but could not find a significant advantage compared to the current form in terms of the final FIDs. Intuitively, the supervised contrastive loss used for $L_{\tt con}^{-}$ already encourages different augmentations of a fake sample (along with other samples as well) to be similar in the normalized embedding space implicitly, so that an explicit loss for this may be redundant.
>
> Hence, the current design is minimal and more beneficial in terms of computational efficiency, as using more and different views in the loss would increase memory consumption and forward computation in training.
>
> We will add respective comments to the revision.
>
> ---
> **Q3. ContraD with "HFlip, Trans"?**
>
> Following the suggestion made by you (and R4), we will update Table 5 in the revision to include an additional ablation of ContraD with a weaker augmentation of "HFlip, Trans": as summarized below, we observe that ContraD is still as good as (or better in terms of IS) a strong baseline of "balanced consistency regularization" (bCR) [Zhao et al., 2020]. The performance degradation compared to "ContraD + SimCLR" is due to that SimCLR requires strong augmentations to learn a good representation [Chen et al., 2020], i.e., ContraD with weaker augmentations learns worse representation in terms of its linear evaluation performance as reported in the following table.
>
> | Method  |   Augment.   | FID $\downarrow$ |  IS $\uparrow$ | Lin. Eval. (%) |
> |:--------|:------------:|:----:|:----:|:--------------:|
> | -       | -            | 26.6 | 7.38 |       -        |
> | bCR	    | HFlip, Trans | 14.0 | 8.35 |       -        |
> | bCR + Aug.  | SimCLR       | 20.6 | 7.44 |       -        |
> | ContraD | SimCLR       | 10.9 | 8.78 |  77.5 $\pm$ 0.20  |
> | ContraD - Aug.  | HFlip, Trans | 13.7 | 8.54 |  72.9 $\pm$ 0.04  |
>
> ---
> **Q4. Discrepancy between "SimCLR" in Table 3 and the original paper [Chen et al., 2020]?**
>
> The discrepancy is because we use different training setups: e.g., we use three popular GAN discriminator architectures, namely SNDCGAN, SNResNet-18, and StyleGAN2 for training (from scratch), while the original SimCLR paper uses ResNet-50 [Chen et al., 2020]. As we put more quantitative efforts in our experiments to improve generation quality of GANs rather than representation quality of SimCLR, we think these prior GAN setups are more valuable to test.
>
> More specifically, as mentioned in Section 4.1, we denote "SimCLR" in Table 3 to indicate an ablation of ContraD when $\lambda_{\tt con} = \lambda_{\tt dis} = 0$, i.e., D is only optimized via $L_{\tt con}^{+}$ (Eq. 6), the SimCLR loss, without using fake samples. We will clarify this in the revision.
>
> ---
> - [Chen et al., 2020] A Simple Framework for Contrastive Learning of Visual Representations, ICML 2020.
> - [Kalantidis et al., 2020] Hard Negative Mixing for Contrastive Learning, NeurIPS 2020.
> - [Robinson et al., 2020] Contrastive Learning with Hard Negative Samples, 2020.
> - [Zhao et al., 2020] Improved Consistency Regularization for GANs, 2020.

---

### Author Response · Authors · 2020-11-20
**Summary of Revisions**

Dear reviewers,

Many thanks again for your constructive feedback to improve our manuscript. We have carefully incorporated your comments into this revision, as summarized in what follows:

- Linear evaluation and transfer learning on ImageNet models (Appendix B; R3)
- Additional ablations (Section 4.2): ContraD + "HFlip, Trans" (R1, R4) / "$h_{\tt r}=h_{\tt f}$" (R4)
- Discussion on the choice of $L_{\tt con}^{-}$ (Section 3.1; R1, R2)
- Improved clarity: Table 3 (R1, R3) / Section 3.2 (R2) / Figure 1
- Improved clarity in mathematical notations, including a typo in Eq. 7 (R4)
- Comments on technical contributions (Section 5; R1, R4)
- Comments on "ContraD is beneficial to SimCLR" (Section 5; R3)
- Minor fixes and comments: "... stabilize GAN …" $\rightarrow$ "... improve GAN ..." (R2) / "gan" $\rightarrow$ "GAN" in the citations (R3) / a typo (R4)

These updates are temporarily highlighted in "red" for your convenience.

If you have time, please check this revised manuscript and our previous response, and let us know if there are any other concerns to be clarified. We will be happy to respond to your further comments during the remainder of the author discussion period.

Best regards,

Authors

---

### Decision · Program_Chairs · 2021-01-07
**Final Decision**

**Decision:**

Accept (Poster)

**Comment:**

This paper aims to improve the training of generative adversarial networks (GANs) by incorporating the principle of contrastive learning into the training of discriminators in GANs. Unlike in an ordinary GAN which seeks to minimize the GAN loss directly, the proposed GAN variant with a contrastive discriminator (ContraD) uses the discriminator network to first learn a contrastive representation from a given set of data augmentations and real/generated examples and then train a discriminator based on the learned contrastive representation. It is noticed that a side effect of such blending is the improvement in contrastive learning as a result of GAN training. The resulting GAN model with a contrastive discriminator is shown to outperform other techniques using data augmentation.

**Strengths:**
  * It proposes a new way of training the discriminators of GANs based on the principle of contrastive learning.
  * The paper is generally well written to articulate the main points that the authors want to convey.
  * The experimental evaluation is well designed and comprehensive.

**Weaknesses:**
  * Even though the proposed learning scheme is novel, the building blocks are based on existing techniques in GAN and contrastive learning.
  * The claim that GAN helps contrastive learning is not fully substantiated.
  * It is claimed in the paper that the proposed contrastive discriminator can lead to much stronger augmentations *without catastrophic forgetting*. However, this “catastrophic forgetting” aspect is not really empirically validated in the experiments.
  * The writing has room for improvement.

Despite its weaknesses, this paper explores a novel direction of training GANs that would be of interest to the research community.